# Programmable mammalian translational modulators by CRISPR-associated proteins

Shunsuke Kawasaki[1,4] ✉, Hiroki Ono[1,2,4], Moe Hirosawa[1,4], Takeru Kuwabara[3], Shunsuke Sumi[1,2], Suji Lee[1,2], Knut Woltjen [1] & Hirohide Saito [1,2] ✉

Translational modulation based on RNA-binding proteins can be used to construct artificial gene circuits, but RNA-binding proteins capable of regulating translation efficiently and orthogonally remain scarce. Here we report CARTRIDGE (Cas-Responsive Translational Regulation Integratable into Diverse Gene control) to repurpose Cas proteins as translational modulators in mammalian cells. We demonstrate that a set of Cas proteins efficiently and orthogonally repress or activate the translation of designed mRNAs that contain a Cas-binding RNA motif in the 5'-UTR. By linking multiple Cas-mediated translational modulators, we designed and built artificial circuits like logic gates, cascades, and half-subtractor circuits. Moreover, we show that various CRISPR-related technologies like anti-CRISPR and split-Cas9 platforms could be similarly repurposed to control translation. Coupling Cas-mediated translational and transcriptional regulation enhanced the complexity of synthetic circuits built by only introducing a few additional elements. Collectively, CARTRIDGE has enormous potential as a versatile molecular toolkit for mammalian synthetic biology.

In all living organisms, highly integrated molecular machinery regulates gene expression at each step of the central dogma. Fine-tuned control of gene expression at multiple layers (e.g., transcription, translation, and post-translation) diversifies cellular identity and coordinates multi-cell systems during development; meaning that even minor disruptions may lead to disease development[1,2]. Imitating such control holds great potential to engineer cells for biotechnology and medical applications.

In recent years, synthetic biology has made momentous advances in programming cellular behaviors by exploiting DNA, RNA, and proteins in artificial gene regulatory networks[3]. Natural genetic switches have been redesigned to create new programmable genetic devices and circuits that mimic silicon-based electronic devices, including band-pass filters[4], Boolean logic gates[5–7], arithmetic circuits[5,8], oscillators[8], and memories[9,10]. Transcriptional regulators, such as zinc finger protein fused with transcription factors (ZF-TFs)[11], transcription activator-like effector (TALE)[12] proteins, Clustered Regularly Interspaced Short Palindromic Repeats and CRISPR-associated (CRISPR-Cas) systems[6], recombinases[13], and meganucleases[14], are the most well-studied and characterized biomodules that have been tested for their composability in artificial networks. Recent studies have also succeeded at exploiting post-translational modules, like proteases and protein-protein interactions, as genetic switches[15,16].

Translational control using RNA-binding proteins (RBPs) is one of the most promising and well-exploited synthetic genetic switches for post-transcriptional circuits[17–22]. RBP-based translational regulators are DNA-encodable and have been coupled with transcriptional regulators to build designer cells with high-computational potential[5]. However, the limited number of available RBPs capable of regulating translation prevents us from designing and implementing post-transcriptional circuits with desired behaviors.

[1]Department of Life Science Frontiers, Center for iPS Cell Research and Application, Kyoto University, 53 Kawahara-cho, Shogoin, Sakyo-ku, Kyoto 606-8507, Japan. [2]Graduate School of Medicine, Kyoto University, Yoshida-Konoe-cho, Sakyo-ku, Kyoto 606-8501, Japan. [3]Faculty of Medicine, Kyoto University, Yoshida-Konoe-cho, Sakyo-ku, Kyoto 606-8501, Japan. [4]These authors contributed equally: Shunsuke Kawasaki, Hiroki Ono, Moe Hirosawa. ✉e-mail: kawasaki.shunsuke.5e@kyoto-u.ac.jp; saitou.hirohide.8a@kyoto-u.ac.jp

We hypothesized that CRISPR-associated (Cas) proteins[23,24] are ideal candidates for generating RBP-mediated translational controllers for the following reasons: (1) Cas proteins are guided by their corresponding crRNAs (CRISPR RNA) or sgRNAs (single guide RNA: chimeric crRNA and trans-activating crRNA (tracrRNA)) and could therefore be repurposed as translational controllers by binding to crRNA or sgRNA sequences embedded in the untranslated regions (UTRs) of messenger RNAs (mRNAs)[25]; (2) many Cas proteins are guaranteed to interact with target crRNA or sgRNA in mammalian cells through their well-documented genome-editing activities; and (3) new Cas proteins are being discovered regularly due to huge efforts in the CRISPR field, which means the available repertoire of potential translational regulators will continue to expand.

Here we present a methodology, CARTRIDGE (Cas-Responsive Translational Regulation Integratable into Diverse Gene control), that repurposes Cas proteins as translational repressors and activators in mammalian cells. We tested a wide range of Cas proteins for their abilities to post-transcriptionally repress protein expression from their target mRNAs with high efficiency (OFF switch). We also used these Cas proteins to activate protein expression (ON switch) by including a "switch-inverting" module that converts translational repressors into activators[26]. Furthermore, we demonstrated that (1) CARTRIDGE can be delivered in either DNA or mRNA formats to control translation of target mRNAs; (2) existing techniques for CRISPR-based genome engineering—including conditional regulation, anti-CRISPR, and split-Cas9—can be repurposed for translational regulation; (3) many Cas proteins show high orthogonality to regulate translation, making it possible to construct an expansive list of post-transcriptional circuits, such as the 60 different AND gates that we have constructed to regulate cell fate; and (4) a single Cas protein can simultaneously regulate both transcription and translation, thus exhibiting the potential of creating multi-layered gene regulatory systems using Cas family proteins while limiting the number of required elements to construct complex circuits.

## Results

### Repurposing CRISPR-associated proteins as translational modulators

We first tested *Streptococcus pyogenes* Cas9 (SpCas9), which is a well-characterized Cas9 protein that has been widely used in genome editing[27–29], for translational modulation. To investigate the translational repression ability of SpCas9, we designed SpCas9-responsive mRNA encoding EGFP (SpCas9-responsive EGFP-OFF switch) by inserting the sgRNA for SpCas9 (referred to as Sp_gRNA) in the 5'-UTR (Fig. 1A left and Supplementary Fig. 1A). We co-transfected this SpCas9-responsive EGFP-OFF switch-expressing plasmid (switch plasmid) and a SpCas9-expressing plasmid (trigger plasmid) into HEK293FT cells along with an iRFP670-expressing plasmid as a reference (reference plasmid) (Supplementary Figs. 1B and C). In principle, translation of the mRNA encoded by the switch plasmid produces the reporter EGFP protein in the absence of SpCas9 proteins (Fig. 1A right, ON state, top), whereas translation is repressed when SpCas9 is available and binds to the switch mRNA by recognizing the Sp_gRNA sequence (OFF state, bottom). To analyze the translational repression ability of SpCas9, we first normalized the reporter expression level (EGFP) by the expression level of our reference protein (iRFP670) to account for variations in transfection efficiency (Normalized Intensity, see Materials and Methods section). Next, we divided the Normalized Intensity of the OFF state by that of the ON state to calculate the Relative Intensity. Finally, we normalized the effects of trigger protein induction itself by dividing the calculated Relative Intensity for each experimental condition using that of the "No gRNA" control condition. We observed that EGFP expression from the switch mRNA with Sp_gRNA in the 5'-UTR was efficiently repressed by SpCas9 induction (more than 80%

repression), when analyzed by flow cytometry and fluorescent microscopy (Fig. 1B, Supplementary Figs. 1D, E, and 2).

Since cleavage of the switch plasmid by the DNA-cleaving activity of SpCas9 may reduce the level of EGFP reporter expression, we also examined this possibility by testing SpCas9 mutants—SpCas9 nickase (D10A), nuclease-null dCas9 (D10A and H840A), and Cas9 without a nuclear localization signal (ΔNLS)[27,28,30]—for their repression activities. These mutants repressed the expression of EGFP with a similar efficiency as wild-type SpCas9 (Supplementary Fig. 3A), indicating that the reporter repression observed was not caused by DNA cleavage by SpCas9. We then examined the effects of AcrIIA4, an anti-CRISPR (Acr) protein that inhibits DNA-binding by SpCas9, on reporter expression[31]. The co-transfection of AcrIIA4 had no significant effects on EGFP inhibition, confirming that the EGFP reduction can be directly attributed to translational repression caused by SpCas9 binding to the target mRNA and is independent of the DNA-binding ability of SpCas9 (Supplementary Fig. 3B).

To confirm that EGFP reduction occurred due to post-transcriptional regulation, we tested whether the SpCas9-mediated EGFP regulation could be achieved by mRNA transfection[18,19]. We investigated the performance of the switch in four different cell lines: HEK293FT, HeLa (human cervical cancer cell), A549 (human lung adenocarcinoma epithelial cell), and iPSCs (human induced pluripotent stem cells) by co-transfecting SpCas9-coding mRNA and EGFP-OFF-switch-coding mRNA. In the presence of SpCas9, EGFP expression from Sp_gRNA-inserted mRNAs was repressed in all cell lines, indicating that the effect was post-transcriptional and independent of cell type (Fig. 1C and Supplementary Fig. 4). In addition, we expressed catalytically dead SpCas9 (dSpCas9) in a doxycycline (Dox)-inducible manner from the AAVS1 safe harbor locus in human iPSCs (Tet-ON dCas9 iPSCs) and demonstrated translational regulation. We induced dSpCas9 expression 4 h before mRNA transfection of our SpCas9-responsive EGFP-OFF switch and analyzed EGFP levels the next day. Dox treatment efficiently repressed EGFP expression from the SpCas9-responsive switch in Tet-ON dCas9 iPSCs (96% repression between with and without Dox), but not in a control iPSC (WT) line (Fig. 1D and Supplementary Fig. 5). The results suggest that Cas9-mediated translational control can be applied in cell lines designed to express Cas9 proteins. Collectively, we concluded that a SpCas9-responsive translational OFF switch can be constructed by introducing Sp_gRNA into the 5'-UTR of mRNA, delivered to a variety of cell types via both DNA or mRNA formats.

### Expanding the library of CRISPR-based translational modulators

Several types of Cas proteins have been identified from diverse species[24,32,33]. To investigate whether our strategy could be applied to other Cas proteins, we designed 41 different types of OFF switches using distinct Cas proteins and their cognate sgRNAs (25 and 41 different types of Cas proteins and sgRNAs, respectively) (Fig. 1E, Supplementary Fig. 2, Supplementary Tables 1 and 2). Interestingly, most of the designed OFF switches repressed reporter EGFP expression by a simple insertion of the reported sgRNAs or crRNAs into the 5'-UTR, in the presence of the trigger plasmid (Fig. 1E). Twenty-two of the 41 switches efficiently repressed EGFP expression by over 80% (for 20 Cas proteins) while 7 switches achieved over 90% repression, as analyzed by flow cytometry. These repression efficiencies were comparable to those achieved by L7Ae, MS2CP, and PP7CP-responsive OFF switches used to design post-transcriptional circuits previously[18,19,21] (98%, 97%, and 80%, respectively, Supplementary Table 2). Some switches (e.g., NmCas9-responsive switches) showed a weak response and required further optimization of the sgRNA sequences (Supplementary Text 1). We also aimed to improve the performance of the OFF switches by examining the effects of copy number or insertion position of the sgRNA in the 5'-UTR using several sgRNAs and Cas proteins (Supplementary Text 1 and Supplementary Figs. 6 and 7). Several Cas-responsive switches contained spacers complementary to an

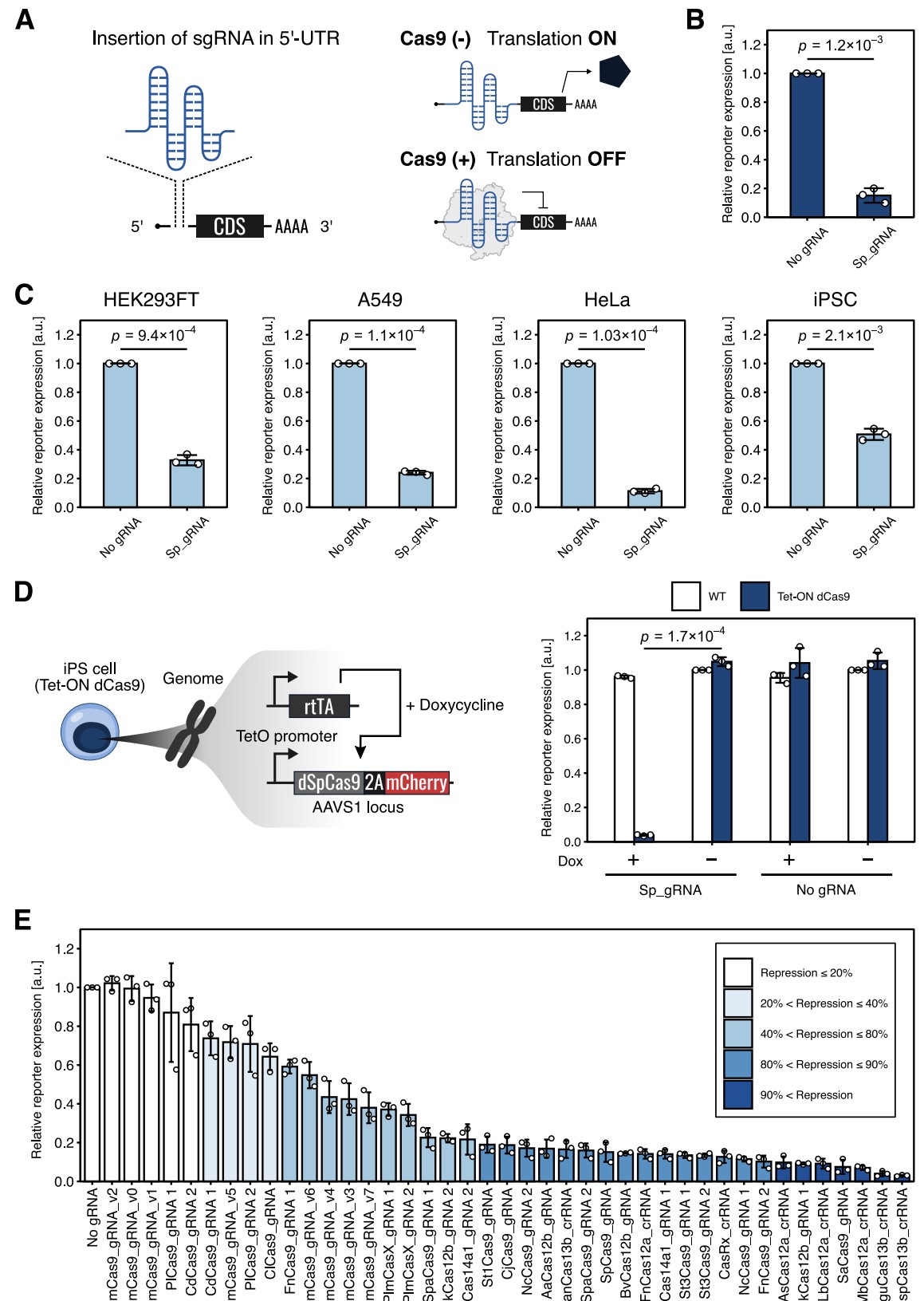

exogenous gene, *Gaussia* luciferase, but we did not observe any clear indications that a spacer sequence was necessary (Supplementary Table 2 and Supplementary Fig. 7). The results suggest that (1) most switches function by simply inserting their reported sgRNAs into the 5'-UTR, and (2) the repression efficiency of Cas-responsive switches could be improved in some cases by optimizing the copy number or insertion position. However, these effects differed depending on the sgRNA used, indicating that further studies are needed to systemically characterize the switch performance of each designed sgRNA-Cas protein pair.

To further expand our arsenal of Cas-mediated translational regulatory switches, we next designed Cas protein-responsive

**Fig. 1 | Cas proteins serve as mammalian translational OFF switches. A** Design of Cas-responsive mRNA OFF switches. sgRNAs or crRNAs are used as protein binding motifs for specific Cas proteins and inserted into the 5′-UTR of target mRNAs. The Cas protein binds to the sgRNA or crRNA region and inhibits the translation of mRNA switches. CDS: coding sequence. **B** Demonstration of the SpCas9-responsive EGFP-OFF switch (Sp_gRNA) as a proof of principle. Relative reporter expressions of the switch are shown. Data were normalized by the value of the control switch without the sgRNA sequence (No gRNA). Data are represented as the mean ± SD from three independent experiments. Statistical analyses were performed using the unpaired two-tailed Student's *t*-test. **C** Relative reporter expressions of the SpCas9-responsive OFF switch in various cell types introduced by mRNA transfection. Translational repression was observed in all cell lines tested. Data are represented as the mean ± SD from three independent experiments. Statistical analyses were performed using the unpaired two-tailed Student's *t*-test. **D** Tet-ON dCas9 human iPSCs with a single copy Dox-inducible catalytically dead dSpCas9 (dCas9 (D10A and H840A)) expression cassette in the AAVS1 locus (left). Relative reporter expressions of the SpCas9-responsive switch introduced by mRNA transfection into Tet-ON dCas9 iPSCs (right). White and blue bar graphs indicate control iPSCs (WT, no dCas9 cassette) and Tet-ON dCas9 iPSCs, respectively. Translational repression was observed only when dCas9 was induced with Dox. Data are represented as the mean ± SD from three independent experiments. Statistical analyses were performed using the unpaired two-tailed Student's *t*-test. **E** Relative reporter expressions of all Cas-responsive mRNA OFF switches tested. Data were normalized to the "No gRNA" control. HEK293FT cells were used in this experiment. Data are represented as the mean ± SD from three independent experiments. a.u. arbitrary unit. Source data are provided as a Source Data file.

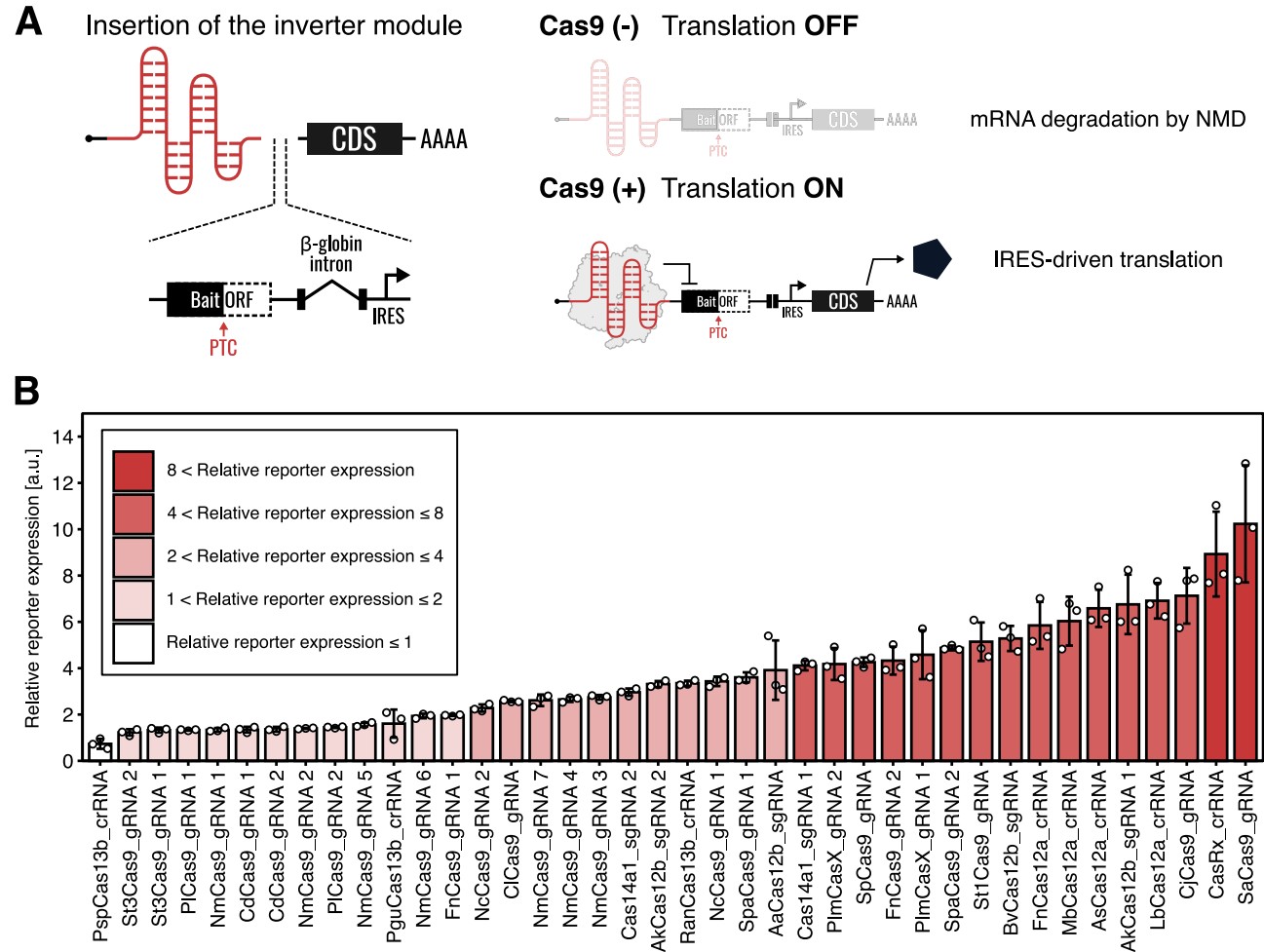

**Fig. 2 | Cas proteins serve as the triggers of RNA-inverters. A** Design of Cas-responsive mRNA ON switches with a switch-inverter module. The inverter module was inserted between the protein binding motifs for specific Cas proteins and CDS. Cas proteins bind to the sgRNA or crRNA and protect mRNA switches from nonsense mediated mRNA decay (NMD). CDS: coding sequence. **B** Relative reporter expressions of all Cas-responsive mRNA ON switches tested. HEK293FT cells were used in this experiment. Data are represented as the mean ± SD from three independent experiments. a.u. arbitrary unit. Source data are provided as a Source Data file.

translational activators (ON switches). We designed a switch-inverting module that converts translational OFF switches to ON switches responsive to Cas proteins (Fig. 2A, left), by controlling mRNA degradation through the non-sense mediated mRNA decay (NMD) pathway[26]. In the absence of Cas proteins, attempted translation of a bait ORF in the designed mRNA terminates at premature termination codons (PTCs) upstream of the exon-exon boundary, consequently leading to NMD activation and mRNA degradation (OFF state, Fig. 2A, top right). In contrast, the binding of Cas proteins to the switch mRNA

blocks the pioneer round of translation of the bait ORF and prevents NMD activation, thus allowing for internal ribosome entry site (IRES)-dependent translation (ON state, Fig. 2A, bottom right). We found that a large majority of Cas proteins functioned as effective triggers of ON switches, with 27 of the 40 tested Cas-responsive switches showing greater than twofold activations (Fig. 2B, and Supplementary Fig. 8). Although the repression efficiency of the PspCas13b-responsive OFF switch was strong, its performance as an ON switch was the weakest among the Cas proteins examined. Some Cas proteins (e.g., Cas12a and

Cas13) have endoribonucleolytic activities and cleave pre-crRNAs to generate mature crRNAs[34,35], making them undesirable for ON switch applications in some cases (Supplementary Fig. 9A). A previous report indicated that switch-inverting modules in principle can be used to convert parental OFF switches into ON switches[26]. Indeed, we observed a moderate correlation between the performance of ON switches and their respective parental OFF switches for most Cas proteins, with the exception of PspCas13b and PguCas13b (Supplementary Fig. 9B; $r = 0.74$). These results support the ON switch mechanism, in which RBPs bind and protect mRNA from degradation to facilitate IRES-mediated translation[26]. Collectively, we show that a wide variety of Cas proteins can serve as both translational repressors and activators through the construction of translational ON and OFF switches. Considering that Cas proteins are well-studied proteins for gene regulation, our strategy could be combined with CRISPR-Cas technologies. Thus, we termed our methodology, CARTRIDGE: Cas-Responsive Translational Regulation Integratable into Diverse Gene control.

## Investigating factors that may affect switch performances

In considering the endoribonucleolytic activity of some Cas proteins, EGFP repression using Cas12a and Cas13b (Fig. 1E) could be attributed to their mRNA cleaving activities. A previous paper also reported a post-transcriptional gene repression system based on the pre-crRNA processing Cas protein, Csy4, suggesting that repression is dependent on Csy4 RNase activity[36]. To examine the dependency of our OFF switches on their RNase activities, we compared translational repression by AsCas12a and its mutant (H800A), which lacks pre-crRNA processing ability[37,38]. Notably, AsCas12a (H800A) also efficiently repressed EGFP expression (81% repression, Supplementary Fig. 10), indicating that, at least for AsCas12a, RNase activity likely did not contribute to the observed EGFP repression. From these findings, we conclude that diverse Cas proteins, including those without RNase activities, can be repurposed as translational repressors.

Next, we searched for potential factors that could affect switch performance (Supplementary Fig. 11). We first analyzed whether structured sgRNAs inserted in the 5'-UTR may affect translation, because several sgRNAs-inserted mRNAs (e.g., NmCas9, NcCas9, AaCas12b-responsive switches) reduced basal protein expression levels even in the absence of the trigger plasmids (Supplementary Fig. 2). Indeed, it is known that rigid secondary structures in the 5'-UTR of mRNAs tend to reduce translation[39]. Thus, we expected that more structured sgRNAs might cause lower basal expression levels of EGFP in ON states (without Cas proteins). We calculated the minimum free energies (MFEs) of sgRNAs and sgRNAs-inserted-5'-UTR (whole 5'-UTR) and calculated the Pearson's coefficient of correlation with basal (ON state) reporter expression levels (Supplementary Fig. 11A). A moderate correlation was observed between reporter expression levels in the ON state and MFEs of the whole 5'-UTR ($r = 0.66$, Supplementary Fig. 11B), indicating that highly structured sgRNAs in mRNA may reduce basal protein expression levels in some cases. Conversely, the correlation between reporter expression levels in the ON state and repression efficiencies was low ($r = 0.24$, Supplementary Fig. 11C, left). Likewise, no obvious correlation was found between repression efficiencies and MFEs ($r = 0.41$, Supplementary Fig. 11C, right). Although the rigidity of an inserted sgRNA may affect the basal expression level, switch performance itself is likely not influenced by the stability of secondary structures of the inserted sgRNAs (MFE) or the basal (ON state) expression of the encoded proteins of interest. Although most switches can be constructed by a simple insertion of the reported sgRNAs, further experiments are required to identify and optimize various parameters to improve the performance of translational switches with sgRNAs of various Cas proteins.

## Repurposing CRISPR technologies for translational regulation

Next, we aimed to conditionally regulate translation using our repurposed CRISPR-based methodology. Conditional regulation of translation with inducers (e.g., small molecules) is useful for fine-tuning gene expression. Recent efforts through RBP engineering have resulted in drug-responsive translational regulators[40,41]. However, it has remained a challenge to develop novel translational regulatory systems because they require in-depth characterization and additional engineering efforts to optimize the individual modulators. A system for translational regulation built upon CRISPR-based technology, instead, would take great advantage of the broad suite of genetic engineering tools dedicated to controlling CRISPR-Cas activity, such as the drug-[42] and light-[43] inducible dimerization of split-Cas9. We expected that such existing knowledge on CRISPR-Cas systems for genome editing and transcriptional regulation could be repurposed easily with CARTRIDGE to create a new powerful system for conditional translational control.

To investigate this possibility, we exploited the split-Cas9 system to develop a conditional translational regulation system. Previous studies reported that SpCas9 can be split into two fragments, composed of residues 1–713 and 714–1368. The divided Cas9 protein fragments can auto-assemble to recover genome editing functionality inside cells[42,44]. Using this split-SpCas9, we constructed a translational NAND gate to repress translation only when both SpCas9 fragments are expressed (Supplementary Fig. 12). Because the repression efficiency of the auto-assembled SpCas9 was low, we fused split inteins from *Rhodothermus marinus* DnaB[45,46] with the split-SpCas9 fragments[44] to improve performance (Fig. 3A). Inteins are intervening protein segments that catalyze protein splicing. DnaB N- and C-inteins fused to residues 1–713 and 714–1368 of SpCas9, respectively, facilitate the intermolecular splicing of the SpCas9 fragments, resulting in the reconstitution of full-length SpCas9. The addition of inteins reinstated translational repression to levels comparable to wild-type SpCas9 and produced clear NAND-like behavior (Fig. 3B).

To develop a chemically controllable, SpCas9-mediated translational regulation system, we combined split-SpCas9 with the iDimerize Inducible Heterodimer System[47], in which separated protein fragments can be assembled by a small molecule (A/C Heterodimerizer) (Fig. 3C). In the presence of A/C Heterodimerizer, the split-SpCas9 fragments assembled and restored translational repression ability. Although the repression efficiency was not as high as that of wild-type SpCas9, translation from the target mRNA with Sp_gRNA was significantly reduced only in the presence of the small molecule (Fig. 3D). Altogether, these data show that our methodology can be combined with existing Cas9 engineering methodologies, such as split-Cas proteins and small molecule-responsive systems, to construct versatile translational regulation systems.

In addition to engineered Cas protein-based systems, natural inhibitors of Cas proteins have been employed to fine-tune the activity of CRISPR-Cas[48,49]. For instance, anti-CRISPR (Acr) proteins are well-studied and promising controllers for Cas protein activity. AcrIIC2 can regulate the genome editing ability of Cas proteins by inhibiting the binding between Cas proteins and their corresponding sgRNAs[50]. We expected this feature to be applicable to CARTRIDGE because AcrIIC2 should block the interaction between Cas proteins and corresponding RNA switches (Fig. 3E), so the effects of AcrIIC2 on Cas9-mediated translational repression were examined by expressing SaCas9 and a SaCas9-responsive OFF switch (Sa_gRNA-EGFP-OFF switch) with or without AcrIIC2. Relative reporter expressions were enhanced by the increasing amount of AcrIIC2-expression plasmid transfected (Fig. 3F), thereby demonstrating the suppression of the SaCas9-sgRNA interaction by AcrIIC2 and translational derepression. As such, Acr proteins can be utilized as a translational controller and, overall, CARTRIDGE can be combined with a variety of artificial and natural CRISPR regulatory systems for translational modulation.

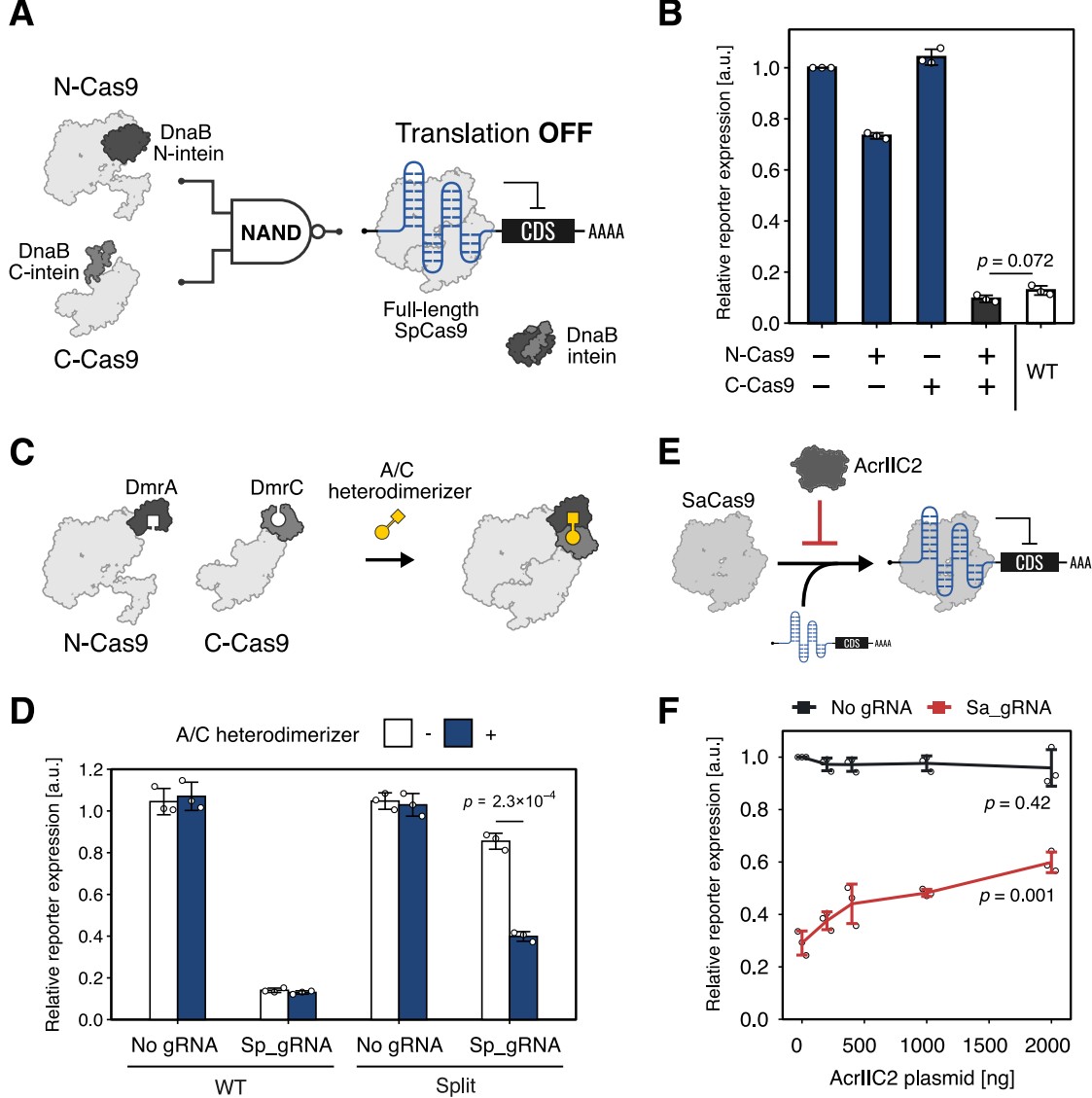

**Fig. 3 | Repurposing CRISPR technologies for translational regulation. A, B** The split-Cas9-based NAND gate. **A** Schematic diagram depicting the NAND gate with intein-fused split-Cas9 proteins. Translational repression occurs only when both protein fragments exist. **B** Result of the reporter assay. Production of the output protein is repressed when both input proteins exist ([+,+]). Relative reporter expressions were normalized to the [−,−] state. **C, D** Drug-inducible translational regulation with split-Cas9. **C** Schematic diagram depicting the split-Cas9 protein fragments fused with the iDimerize system. **D** Result of the reporter assay. Split-Cas9 repressed translation in the presence of 500 nM A/C heterodimerizer, while wild-type Cas9 (WT) constitutively repressed translation of the switch. Relative reporter expressions were normalized to the control condition, in which "No gRNA" and "No trigger" (control trigger) plasmids were co-transfected. **E, F** Translational regulation by an anti-CRISPR protein. **E** Schematic diagram depicting AcrIIC2-mediated translational regulation. AcrIIC2 inhibits binding between SaCas9 protein and its sgRNA. **F** Relative reporter expression of the "No gRNA" control or SaCas9-responsive switch (Sa_gRNA) co-transfected with increasing amounts of AcrIIC2-expression plasmid. Relative reporter expressions were normalized by the value at 0 ng AcrIIC2. *P*-values shown correspond to comparisons between 0 ng and 2000 ng. HEK293FT cells were used in these experiments. Data are represented as the mean ± SD from three independent experiments. Statistical analyses were performed using the unpaired two-tailed Student's *t*-test. a.u. arbitrary unit. Source data are provided as a Source Data file.

## The orthogonality of Cas-responsive switches

An expanded repertoire of Cas proteins-mediated translational regulators should provide a means to scale up synthetic post-transcriptional networks. Orthogonality (specificity in the interaction between the RNA and RBP) is a key criterion for genetic circuitry used when building complex circuits. To investigate the reliability of Cas-responsive switches in complex circuits, we verified the orthogonality of OFF and ON switches that responded to 25 different types of Cas proteins (Fig. 4A, Supplementary Figs. 13 and 14). We transfected all combinations of switch and trigger plasmids into HEK293FT cells and captured fluorescent images for detailed analysis. In summary, we observed distinct crosstalk within the Cas12a family and several pairings involving Cas9, Cas12b, or Cas13b family members to also possess low specificities, together indicating that these pairs should be excluded from further analysis for the identification of reliable orthogonal combinations.

To identify optimal orthogonal combinations of Cas proteins and switches, we defined the orthogonality of a combination as the minimum cosine distance between the fold change profiles of any two Cas proteins in the combination. Using this metric, we screened all the possible combinations and determined the best combination for each size. (Fig. 4B). A calculated distance close to 1 indicates a combination

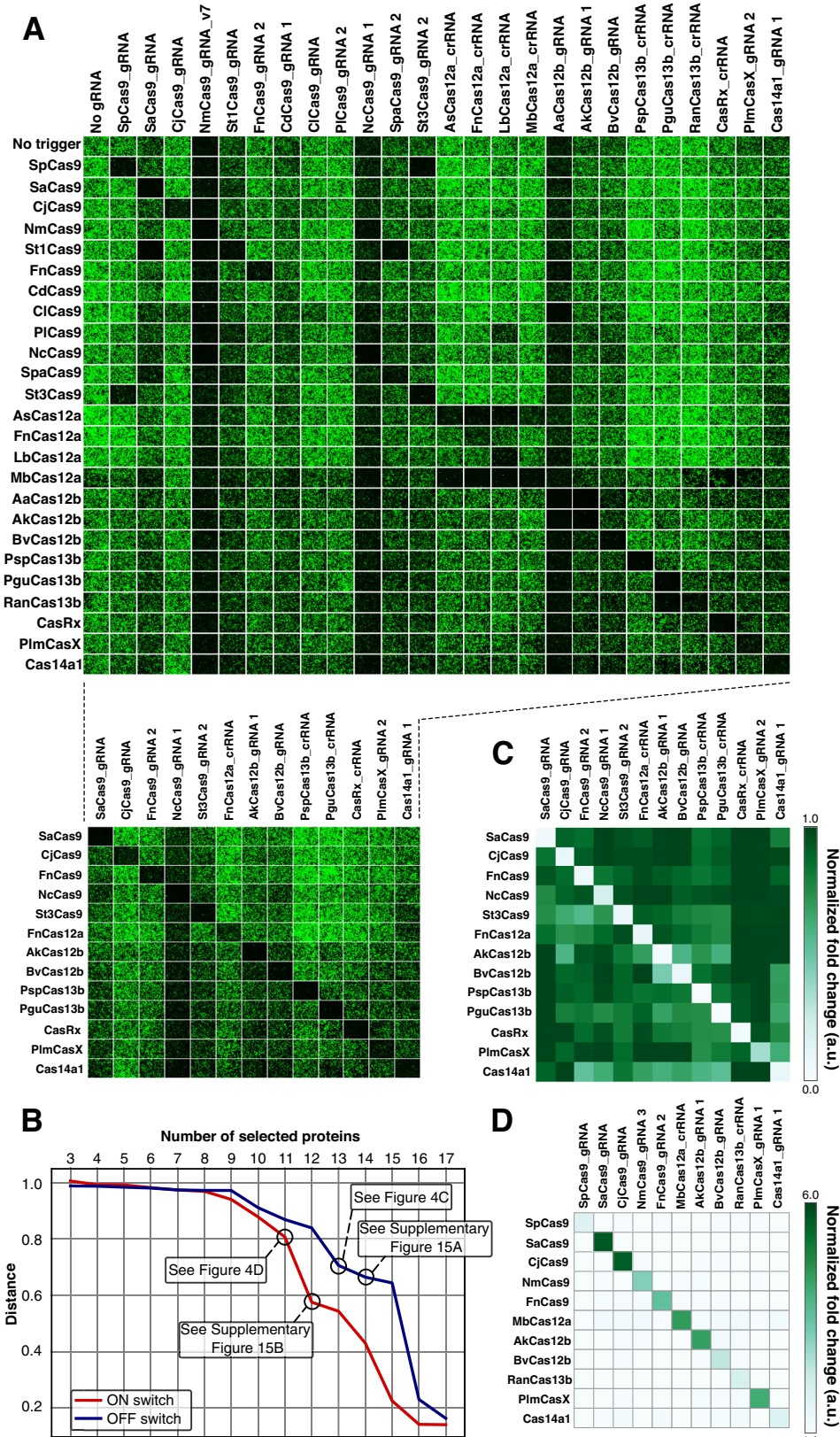

**Fig. 4 | Orthogonality among representative Cas-responsive switches.**
**A** Fluorescent cell images of a 25 × 25 orthogonality matrix of representative Cas proteins and Cas-responsive switches (top). HEK293FT cells were used in these experiments. Mutually orthogonal pairs are shown (bottom). Pruned images on the bottom indicate mutually orthogonal sets. **B** Distribution of the best cosine distance in the different numbers of Cas protein sets. Cosine distances of all the combinations of Cas proteins in the set sizes N = 3-17 were calculated.

Combinations which are lower than the threshold of 0.7 as OFF and ON switches were indicated in Supplementary Fig. 15A and B, respectively. **C, D** Estimated crosstalk between mutually orthogonal pairs in OFF switches (**C**) and ON switches (**D**). Data were obtained from imaging analysis. The heatmap shows the mean values from three independent experiments performed on different days. a.u. arbitrary unit. Source data are provided as a Source Data file.

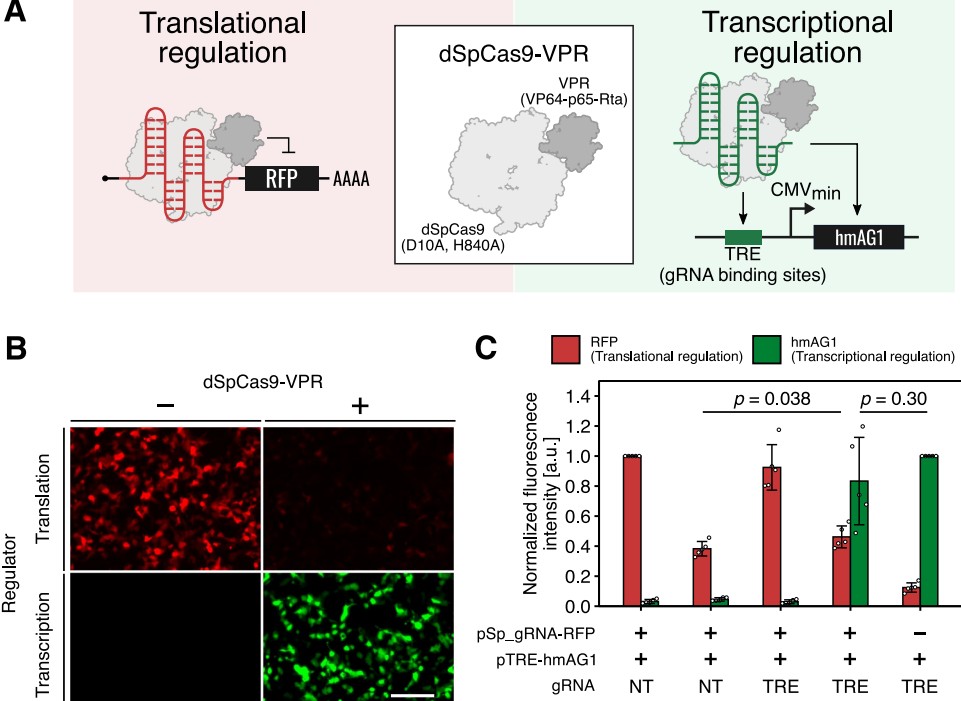

**Fig. 5 | Simultaneous regulation of transcription and translation with transcription-factor fused, catalytically dead SpCas9. A** Schematic diagram of the simultaneous regulation system. dSpCas9-VPR (catalytically dead SpCas9 fused with a transcriptional activator) was used as the trigger protein. Translational repression and transcriptional activation were monitored with TagRFP and hmAG1, respectively. Tet-responsive elements (TRE) were used as the sgRNA binding site. **B** Fluorescence microscopy images captured for each condition. Translational repression and transcriptional activation were observed simultaneously when dSpCas9-VPR was transfected. Scale bar, 200 μm. **C** Quantitative data of reporter expression levels. The data were obtained by imaging analysis. TagBFP was co-transfected as a reference and used to define the transfection-positive area. pSp_gRNA-TagRFP: plasmid for expressing TagRFP, whose translation was regulated by SpCas9. pTRE-hmAG1 plasmid for expressing hmAG1, whose transcription was regulated by Tet-responsive promoter. HEK293FT cells were used in this experiment. TRE: TRE-targeting sgRNA. NT: non-targeting sgRNA. Data are represented as the mean ± SD from 5 independent experiments. Statistical analyses were performed using the unpaired two-tailed Student's *t*-test. a.u. arbitrary unit. Source data are provided as a Source Data file.

that behaves with ideal orthogonality, whereas low distance values highlight combinations with potential crosstalk. In principle, as the combination size grows, the cosine distance of the best combination decreases because a Cas protein with low specificity must eventually be included in the combination (Fig. 4B). Among the OFF switches tested, we identified thirteen different Cas proteins that maintained a relatively large distance value to selectively repress the translation of target mRNAs (Fig. 4C), whereas fourteen Cas protein combinations including FnCas12a and LbCas12a proteins exhibited substantial crosstalk (Supplementary Fig. 15A). We thus concluded that the thirteen Cas proteins shown in Fig. 4C can maintain orthogonality. For ON switches, we found eleven different Cas proteins (Fig. 4D) to be orthogonal because the distance value sharply declined when twelve Cas proteins (including FnCas12a and LbCas12a) were included (Supplementary Fig. 15B).

The pairs which showed crosstalk seemed to be close within specific Cas families. As mentioned, the Cas12a family and several pairs in the Cas9, Cas12b, or Cas13b families were characterized by low orthogonality. The frequent crosstalk between the Cas12a family is likely due to their highly conserved crRNA features, consistent with findings from a previous study[51]. Indeed, hierarchical clustering of sgRNAs revealed that crosstalked Cas9 families (e.g., SpCas9-St3Cas9, St1Cas9-SaCas9) are grouped in close clusters (Supplementary Fig. 16), suggesting that sequence similarity between sgRNAs may be an important contributing factor to crosstalk of translational repression by Cas9 families. Nevertheless, various Cas proteins showed orthogonality as both OFF and ON switches, indicating that we successfully expanded the repertoire of composable translational circuitry with orthogonal Cas protein modules.

## Dual regulations by a single Cas protein

Cas proteins are used to regulate transcription by fusing transcription modulators like VPR[51] or KRAB[52] to either enhance or suppress, respectively, transcription at sites recognized by supplied sgRNA sequences. It should thus be possible to use CARTRIDGE to enable simultaneous control of transcription and translation (e.g., activating transcription while repressing translation). To achieve this, we used defective SpCas9 (dSpCas9) fused with a transcriptional activator, VPR, as a trigger protein (dSpCas9-VPR, Fig. 5A). As expected, this fusion protein activated the transcription of hmAG1 guided by co-transfected Tet-responsive elements (TRE)-targeting sgRNA (transcription ON), while decreasing the translation via a SpCas9-responsive OFF switch (encoding TagRFP as a reporter: referred to as Sp_gRNA-RFP) (translation OFF) (Fig. 5A, B). There was no significant difference between the hmAG1 intensity in the absence and presence of Sp_gRNA-RFP but a slight difference between the RFP intensity in the absence and presence of TRE-targeting sgRNA, indicating that there is limited competition between sgRNA and Sp_gRNA-RFP for dSpCas9-VPR (Fig. 5C). Thus, CARTRIDGE can be used for dual-functional regulation (i.e., regulation at both translation and transcription levels).

Transcriptional regulatory circuits often require independent triggers for transcriptional activation and repression. In contrast, one of the advantages of CARTRIDGE is that the same trigger can serve as both a repressor and an activator. This feature of the translational regulatory system helps to minimize the size of gene circuits. A recent study demonstrated that a trigger in translational regulation can target multiple translational switches, referred to as "fan-out" ability[53], suggesting that it is also possible to regulate translational repression and activation simultaneously by a single Cas protein. To examine this

possibility, we co-transfected translational OFF (TagRFP) and ON (EGFP) switches designed to respond to SaCas9 with a trigger and the reference (iRFP670) plasmid (Supplementary Fig. 17A). As expected, we observed that CARTRIDGE controlled translational activation and repression in a fan-out manner (Supplementary Fig. 17B, C).

## Validating the composability of Cas-responsive RNA switches

To verify the composability of Cas-responsive OFF switches in multi-layered translational circuits, we explored the AND gate circuit as a representative 2-input multi-layered logic circuit (Fig. 6). An AND circuit expresses the output only in the presence of both inputs (input pattern [1,1]). Two-input translational AND circuits can be constructed using 3 types of OFF switches (Fig. 6A). We arbitrarily chose 6 Cas-responsive switches and constructed mediator plasmids that express Cas protein C except when both Cas proteins A and B are present. Accordingly, we prepared and tested 60 unique AND circuits generated from different triple combinations of 6 available Cas-responsive switches, thus representing the largest number of translational AND gates ever tested (Fig. 6B and C and Supplementary Table 3). To measure the similarity between the intended function of the AND gates and experimental data, we calculated the net fold-change (ratio of the mean fluorescence intensity of the ON state ([1,1] state) and OFF states ([0,0], [1,0], and [0,1])) and the cosine similarity, which are used as a metric to evaluate genetic circuits[13]. Twenty-one circuits showed angles less than or equal to 20°, which is comparable to previously constructed translational AND circuits[18] (Fig. 6D). All circuits that included the NcCas9-responsive switch showed angles over 20°, perhaps due to low basal expression of the mediator proteins and in turn did not adequately repress reporter expression (Fig. 6C and Supplementary Table 3).

Next, we investigated whether the AND gates' performance was sufficient to control cell phenotype. The circuit was designed to induce the expression of a pro-apoptotic protein, hBax, only when both input Cas proteins are present (Fig. 6E). Among 60 patterns of AND gates, we chose two circuits that showed a good performance: one is AND gate #2, which uses PguCas13b and SaCas9 as inputs and PspCas13b as the mediator; and another is AND gate #43, which uses PspCas13b and SaCas9 as inputs and AkCas12b as the mediator (Supplementary Table 3). Apoptotic and dead cells significantly increased only when both inputs were introduced in these circuits ([1,1] state), whereas no obvious changes in cell viability were observed in the OFF states ([0,0], [1,0], and [0,1]) and controls (Fig. 6F, Supplementary Fig. 18, and Supplementary Table 4). The percentages of apoptotic and dead cells in [1,1] states were comparable to when we transfected only the hBax plasmid into 293FT cells. Thus, CARTRIDGE-AND gates can regulate the translation of a pro-apoptotic gene and control cell death by considering input from two Cas proteins.

In addition, we tested the composability of Cas-responsive switches in other multi-layered circuits (Fig. 7). The 4-stage repression cascade was constructed with MbCas12a, SaCas9, PguCas13b, and PspCas13b-responsive switches (Fig. 7A). Although expression levels of the reporter were gradually reduced as we increased the layer of translational control, clear transitions between the signals (ON to OFF, and OFF to ON) were observed (Fig. 7B). We also evaluated the performance of layered circuits composed of two tandemly interconnected ON switches (Fig. 7C). We constructed two circuits, one used SaCas9 and CjCas9 as a trigger and a mediator, respectively, and the other had the order of the two proteins reversed (i.e., CjCas9 and SaCas9). The former circuit highly expressed the reporter only when the trigger, mediator, and reporter plasmids were all transfected (Fig. 7D). However, the latter circuit showed leaky reporter expression in the absence of a trigger, although the inclusion of the trigger further upregulated output reporter expression (see (trigger, mediator, reporter) = (−,+,+) and (+,+,+) conditions in Fig. 7D). A possible explanation for this is that the SaCas9-responsive ON switch is highly

sensitive and leaky expression of SaCas9 from the mediator plasmid may have induced reporter expression slightly. Furthermore, we constructed a NOR gate by tandemly inserting two different crRNAs in a single mRNA (Fig. 7E). A NOR gate should express reporter protein only in the absence of both inputs. We used PspCas13b and PguCas13b as representative inputs and designed two different NOR gate constructs: one with PspCas13b_crRNA on the 5' side, and another has it on the 3' side. In both cases, high reporter expression was observed only in the absence of both inputs as expected (Fig. 7F). Taken together, synthetic gene circuits created using CARTRIDGE can implement multiple logic operations in living mammalian cells and control their phenotypes.

## An arithmetic circuit via compacted modules

Thus far, we have demonstrated CARTRIDGE to provide the genetic circuitry toolbox with high orthogonality, dual-functionality, and composability. We next attempted to build a more complex circuit, an arithmetic circuit that has higher-order synthetic networks with complex signal wiring, using Cas proteins and corresponding sgRNAs. Previous transcription-translation coupled binary arithmetic circuits required four genetic modules, two transcriptional controllers and two translational controllers[5]. We attempted to build a binary arithmetic circuit with fewer modules to demonstrate this advantage because CARTRIDGE can simultaneously control transcription and translation using a single Cas protein (Fig. 5). We designed a half-subtractor with two catalytically dead Cas proteins, dSpCas9 and dSaCas9, fused with VPR as inputs (Fig. 8). Both proteins were used as transcriptional regulators and operated in an orthogonal manner to each other. To monitor the calculated difference (D) and the borrow (Bo), we chose the fluorescent proteins hmAG1 and TagBFP as our output signals. dSaCas9-VPR and dSpCas9-VPR activated the transcription of Sp_gRNA-TagBFP and Sa_gRNA-TagBFP mRNA, respectively, and the translation of these mRNAs was orthogonally suppressed by dSpCas9-VPR and dSaCas9-VPR. TagBFP expression is therefore regulated in an XOR manner (outputs are produced only when exactly one Cas protein is available (Fig. 8A, TagBFP). Meanwhile, the expression of hmAG1 is regulated in a dSpCas9 NIMPLY dSaCas9 manner: hmAG1 is produced only when dSpCas9 alone is present. To achieve this logic, the sgRNA is quarried out from the 3'-UTR of the Sa_gRNA-TagBFP transcript by a hammerhead ribozyme (HHR) and a hepatitis delta virus ribozyme (HDVR) and guides dSpCas9-VPR to the promoter region of hmAG1[54,55]. We also inserted a MALAT1 triplex upstream of the ribozymes to stabilize the transcript after the loss of the poly-A tail[56]. The 5'-UTR of the hmAG1 transcript contains SaCas9 sgRNA, which is translationally repressed by dSaCas9. Thus, dSpCas9 activates the transcription of hmAG1, but its translation is suppressed by dSaCas9 (Fig. 8A, hmAG1). Imaging analysis indicated the designed half-subtractor behaved as expected (Fig. 8B). Thus, CARTRIDGE can implement complex logic operations with improved compactness and stage programmability in mammalian cells.

## Discussion

In this study, we developed CARTRIDGE to repurpose Cas proteins into translational repressors and activators. We initially tested 25 Cas proteins and found that 20 Cas protein-responsive OFF switches were capable of >80% translational repression. Among these, 14 switches were constructed by a simple insertion of reported sgRNA at the 5'-UTR, and the other 6 switches were further developed by optimizing the sgRNA sequences. We converted these Cas-responsive OFF switches to ON switches by using a switch-inverting module[26] and generated 27 types of ON switches. Furthermore, we designed 24 different orthogonal regulators (13 OFF and 11 ON switches each), multi-layered circuits including 60 translational logic AND gates, and constructed a complex arithmetic circuit using compacted modules. To our knowledge, this is the largest number of translational regulatory

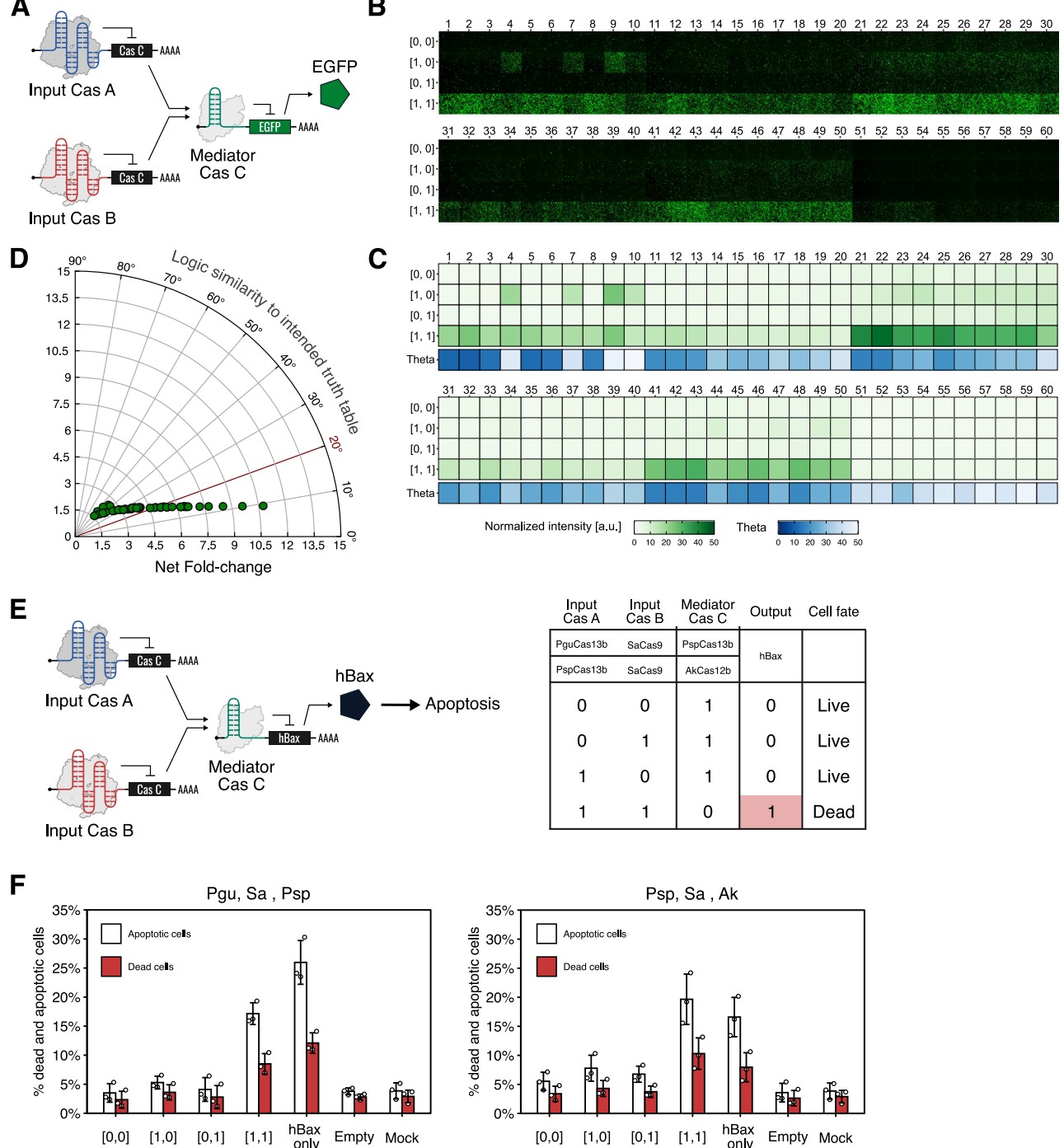

**Fig. 6 | Multi-layered translational circuits with Cas-responsive switches.**
**A**–**D** Combinatorial validation of Cas-responsive switches' composability. Sixty distinct 2-input translational AND gate circuits were tested in HEK293FT cells. **A** Schematic diagram of the 2-input AND gate composed of three Cas-responsive switches. Six Cas proteins were selected and arranged as Cas proteins A, B (as inputs), and C (as a mediator). Thus, a total of 60 circuits were tested. **B** Fluorescence microscopy images of the 60 distinct 2-input AND gates with four input patterns. The absence and presence of the inputs are indicated by 0 and 1, respectively. The ideal AND gate shows reporter expression only in the [1,1] state. **C** Heatmap of GFP expression levels and vector proximity angles. Data were obtained by flow cytometry analysis. **D** A polar coordinate plot of the net fold-change and vector proximity angles. Data are represented as the mean from three independent experiments. **E** Schematic diagram depicting the cell death-inducing AND gate (left) and its truth table (right). When both two input Cas proteins exist, cells are dead due to hBax expression. **F** Percentages of dead and apoptotic cells. The left graph indicates the AND gate consisting of PguCas13b and SaCas9 as inputs and PspCas13b as a mediator. The right graph shows the AND gate composed of PspCas13b and SaCas9 as inputs and AkCas12b as a mediator. HEK293FT cells were used in this experiment. Data are represented as the mean ± SD from three independent experiments. a.u. arbitrary unit. Source data are provided as a Source Data file.

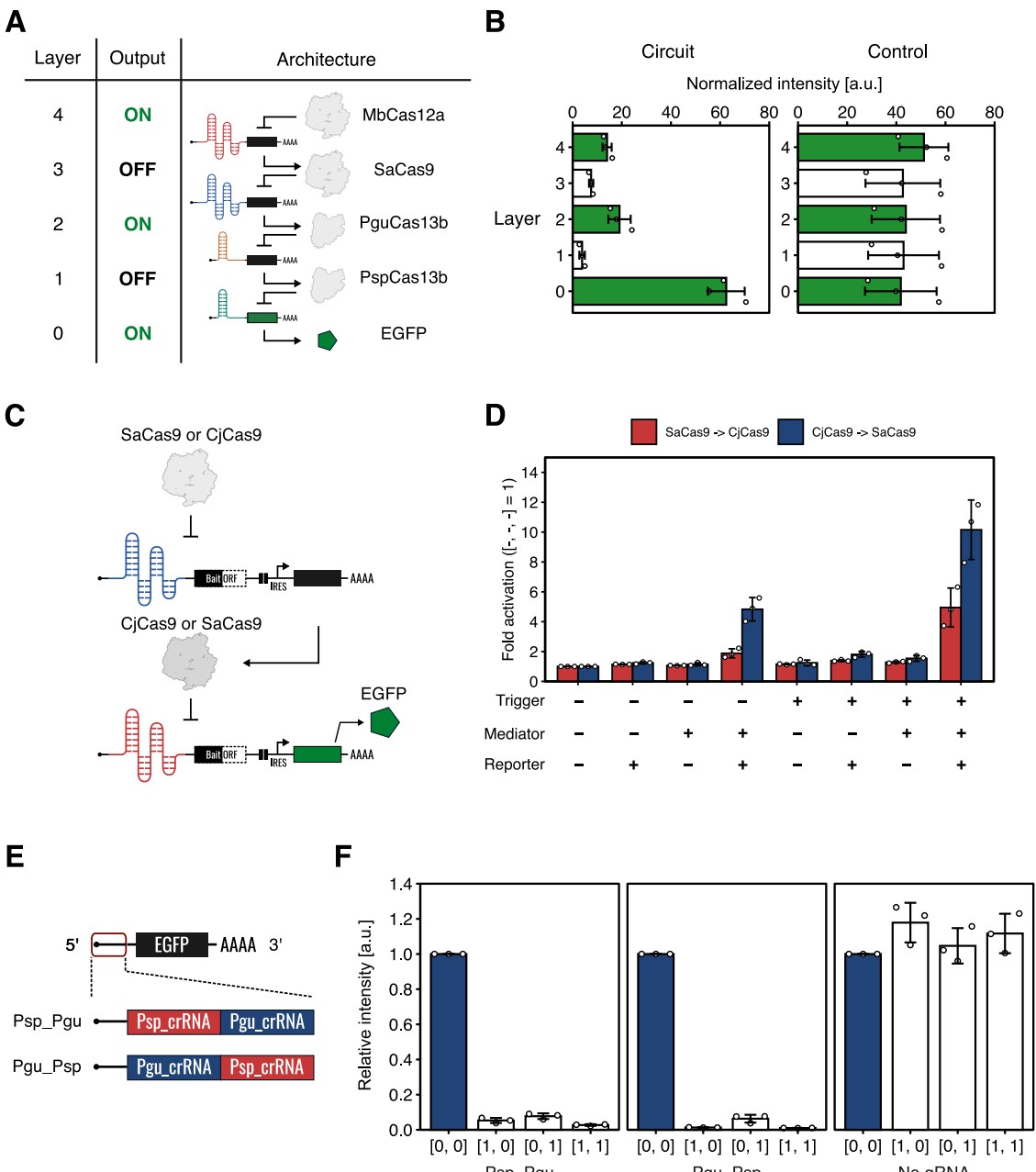

**Fig. 7 | Performance of Cascade circuits and NOR gate. A** Schematic diagram depicting the four-stage repression cascade. MbCas12a, SaCas9, PguCas13b, and PspCas13b-responsive OFF switches were used. **B** Normalized EGFP intensity of the circuits. Signal transitions (i.e., ON to OFF and OFF to ON) were observed in the circuit (left), not control (right). Error bars represent mean ± SD from three independent experiments. **C** Schematic diagram depicting the layered circuits composed of two tandemly interconnected ON switches. SaCas9- and CjCas9-responsive ON switches were used. SaCas9 (or CjCas9) and CjCas9 (or SaCas9) were used as triggers and mediators, respectively. **D** Fold activation of the circuit. Red bars indicate data from experiments in which SaCas9 and CjCas9 were used as trigger and mediator, respectively; whereas blue bars represent data from experiments in which they were reversed as trigger (CjCas9) and mediator (SaCas9). A high expression level of the reporter was observed when all the components were co-transfected. Error bars represent mean ± SD from three independent experiments. **E** Schematic diagram depicting the NOR gate. The crRNAs of PspCas13b and PguCas13b were inserted in tandem into a single mRNA. Psp_Pgu indicates that PspCas13b_crRNA and PguCas13b_crRNA were placed at the 5′ side and the 3′ side in 5′-UTR, respectively. Pgu_Psp shows that PguCas13b_crRNA and PspCas13b_crRNA were placed at the 5′ side and the 3′ side in 5′-UTR, respectively. **F** Relative intensity of the NOR gate. High reporter expression was only observed when both inputs were absent. HEK293FT cells were used in these experiments. Error bars represent mean ± SD from three independent experiments. a.u. arbitrary unit. Source data are provided as a Source Data file.

devices and translational logic gates ever tested, since only a limited number of translational regulatory proteins have been used to date to build post-transcriptional circuits. Importantly, our CARTRIDGE strategy, in which Cas proteins are used as translational modulators, is compatible with other conventional CRISPR technologies. For example, the use of split-Cas9 and anti-CRISPR proteins enabled us to perform small molecule-based and conditional translational regulation (Fig. 3), suggesting that transcriptional and genome regulatory systems for CRISPR can be combined with CARTRIDGE. Together, our findings highlight the feasibility of using CARTRIDGE to incorporate an assortment of promising modules to construct multi-layered synthetic gene circuits.

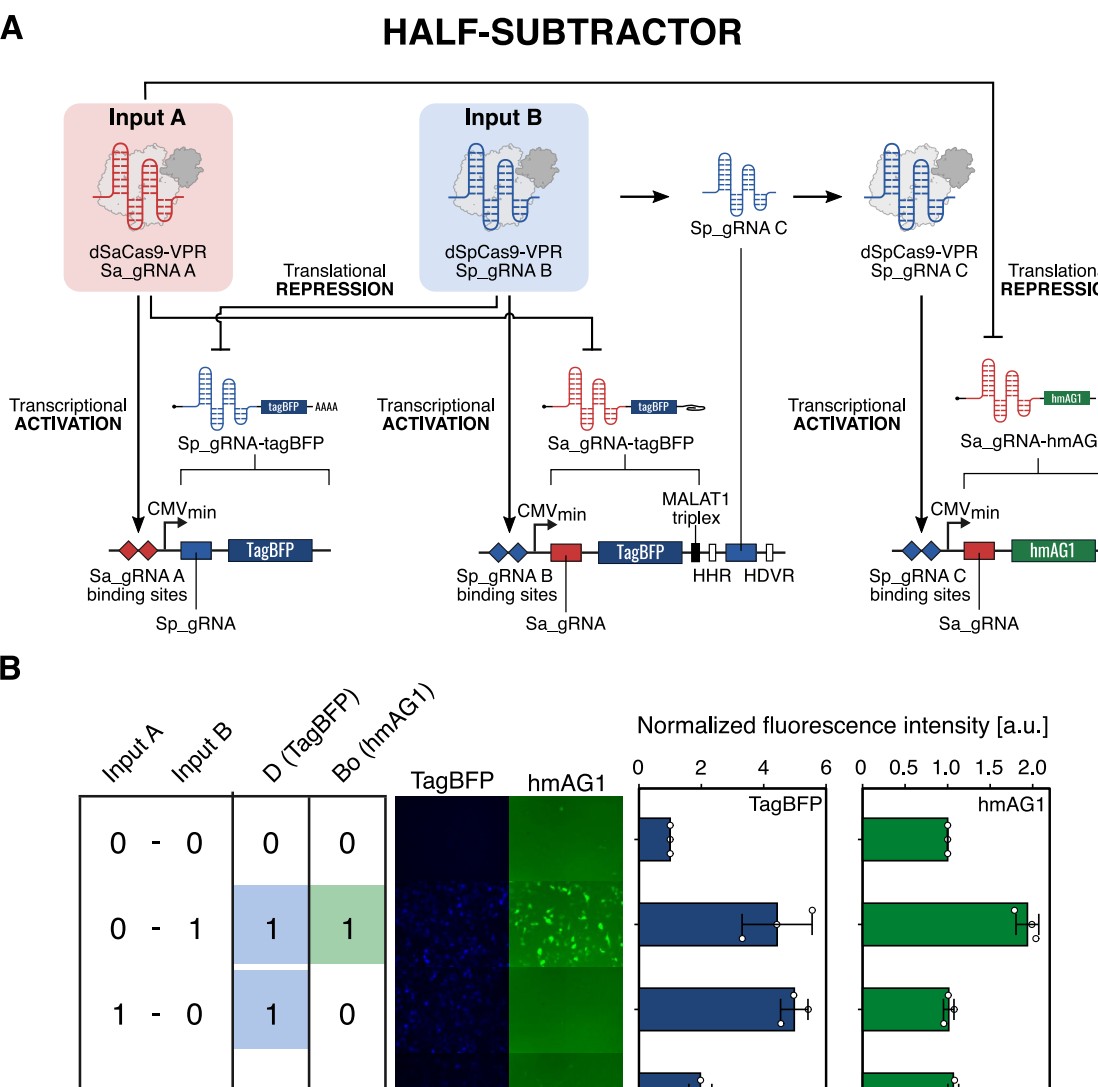

**Fig. 8 | Design and processing performance of a half-subtractor with fewer bio-modules. A** Schematic diagram depicting the 2-input half-subtractor. VPR-fused dSpCas9 and dSaCas9 were used as the initial inputs (top left). Transcription of mRNA switches (middle) from the corresponding plasmids (bottom). Translation is regulated by the indicated input dCas9. **B** Fluorescence microscopy images and analysis of the half-subtractor performing arithmetic subtraction of the SpCas9 input from the SaCas9 input by calculating the difference, D (TagBFP), and borrow, Bo (hmAG1). HEK293FT cells were used in this experiment. Data are represented as the mean ± SD from three independent experiments. Scale bar, 200 μm. a.u. arbitrary unit. Source data are provided as a Source Data file.

Recent efforts have expanded the catalog of available synthetic toolkits for constructing gene circuits. The recombinase-based approach BLADE[13] provides highly robust circuits. However, it is an irreversible operation, and it is difficult to reset the cell state. COMET[11], which uses ZF-TFs as the transcriptional regulator, and a similar approach that uses meganuclease[14], can perform reversible regulation in principle. These transcriptional regulation strategies, however, cannot be adapted to DNA-free transgene methods like mRNA delivery. In contrast, CARTRIDGE can be introduced into cells to drive synthetic circuits at the translation level via mRNA transfection (Fig. 1D and Supplementary Fig. 4). Furthermore, CARTRIDGE is compatible with transcriptional circuits so that more complex genetic circuits can be built with fewer bio-modules as building blocks (Fig. 8). Recent studies have also constructed post-translational circuits by exploiting proteases[16], coiled-coil domains[15], and de novo designed heterodimeric domains[57]. Given that CARTRIDGE is an RBP-based regulatory system, these engineering approaches can likely be incorporated to make even more complex circuits. Indeed, one study has already tested Cas9 with such programmable features[58].

During this study, several groups reported post-transcriptional regulation using Cas proteins. Among Cas proteins, two studies used only Cas12a and mainly focused on how to supply crRNA from mRNA, which encodes Cas12a itself[37,59]. These studies tested several Cas12a variants, but most exhibit strong crosstalk[60] (Fig. 4) that would likely hinder large-scale assembly of bio-modules. Another study reported a novel post-transcriptional regulation system, PERSIST, which uses Cas proteins that have endoribonuclease activity[61]. Although this system greatly expanded strategies for synthetic post-transcriptional regulation, its reliance on mRNA degradation by endonucleases limits the range of Cas proteins that it can use to only those with RNase activity. CARTRIDGE, in contrast, does not have such restrictions and therefore can simultaneously employ a variety of Cas proteins, enabling the construction of gene circuits with far greater complexity. Another study also reported a translational activation system with SpCas9[62], but limited its application to yeast and required extensive optimization

of the sgRNA insertion position. In contrast, we validated the properties of many more Cas proteins (25 Cas proteins investigated as translational regulators) in mammalian cells, thus illustrating the potential power of CARTRIDGE and its compatibility with various CRISPR-related technologies.

From our findings, we believe that CARTRIDGE offers several advantages for the design and construction of gene circuits. (1) As more and more CRISPR-associated genes are being discovered through extensive efforts from the genome editing field, an ever-expanding list of Cas proteins will become available for us to use as translational modulators. (2) CARTRIDGE can be combined with both existing and emerging CRISPR technology. In addition to the three translational regulation methods illustrated in this study (Fig. 3; split-Cas9, drug-inducible regulation, and Acr), other possible approaches include light-inducible regulation[43], protease-combined regulation[58], CRISPR-based gene circuits[6], and molecular event recorders[10]. These regulatory mechanisms will allow us to finely tune and conditionally control transgene expression at both transcriptional and post-transcriptional levels. (3) CARTRIDGE enables simultaneous regulation of transcription and translation using a single Cas protein (Figs. 5 and 8), thereby reducing redundancy when constructing gene circuits. This property grants CARTRIDGE the potential to achieve higher information density and processing power with fewer bio-modules (Fig. 8). CARTRIDGE can therefore be the basis of "biological integrated circuits" to facilitate the development of easily designable biocomputers. Due to this same feature, the metabolic burden in the chassis will likely be minimized[63]. (4) CARTRIDGE provides the opportunity to study the design principles of protein-responsive mRNA OFF and ON switches. In this study, we developed a library of Cas-responsive switches as a wide variety of translational modulators. Further elucidation of the relationship between translatability and biochemical and biophysical parameters (e.g., structural features of mRNA and binding kinetics) may lead to improved performance of mRNA switches. (5) CaRTRIDGE is adaptable for delivery both as DNA or synthetic mRNA into various cell types (e.g., human iPSCs) (Fig. 1C, D) and various cell lines engineered to express different Cas variants and Cas fusion proteins.

In addition, CARTRIDGE has the potential to contribute back to the genome editing field. For example, our system can be employed as an initial screening platform for identifying optimal Cas protein-sgRNA pairs in human cells. Cas9 proteins with inefficient translational repression (e.g., ClCas9-ClCas9_gRNA and FnCas9-FnCas9_gRNA 1) also seem to lack genome editing ability in mammalian cells[64,65]. Thus, CARTRIDGE may be used to exclude the use of certain Cas protein-sgRNA pairs that are inactive in cells for genome editing. Furthermore, some small molecules and anti-CRISPR proteins are reported to disrupt Cas9-DNA interactions and influence Cas9 genome editing ability[48,66]. CARTRIDGE may therefore facilitate the discovery of these inhibitors against Cas protein-sgRNA complexes to provide more powerful genome editing regulators.

A major drawback of CARTRIDGE is the gene size of Cas proteins, which may limit its use for some applications. In the present study, we showed the applicability of split-Cas9 to reduce the size of plasmids used (Fig. 3), but a more efficient gene delivery system free of size limitations will no doubt be helpful. Another potential solution is the use of smaller Cas proteins. Our data suggest that the ability to mediate translational control is independent of Cas protein size (Supplementary Table 1). CARTRIDGE may thus directly benefit from future discoveries of smaller Cas proteins[67,68]. Because several RBPs used as synthetic mRNA switches can control translation naturally[69], we anticipate that some Cas proteins may control translation by directly interacting with mRNAs in prokaryotes. In conclusion, we believe that CARTRIDGE will inspire not only RNA- and RBP-based synthetic biology, but also fuel future genome editing technologies and CRISPR studies.

## Methods

### Construction of plasmids

For constructing the switch plasmids, pAptamerCassette-EGFP (available at Addgene, #140288) was digested by AgeI and BamHI. Single-stranded oligo DNAs were annealed to generate double-stranded DNA and then ligated with the digested plasmid. Links to sequences of all the switch plasmids generated are listed in Supplementary Table 5.

For constructing the trigger plasmids, the open reading frame (ORF) of trigger proteins was amplified by PCR using appropriate primers. The amplicons were digested by restriction enzymes and then inserted downstream of the CMV promoter of pcDNA3.1-myc-HisA (Invitrogen, #V80020). Links to sequences of all the trigger plasmids generated are listed in Supplementary Table 5.

To construct split-Cas9, hCas9 (obtained from Addgene, #41815) was used as the template. N_Cas9 and C_Cas9 were amplified by inverse PCR using appropriate primers. To generate drug-responsive Cas9, DmrA and DmrC were amplified and then fused to each split-Cas9 (N_Cas9 and C_Cas9) using the In-Fusion HD Cloning Kit (Clontech, #639634). Links to sequences of all the plasmids generated are listed in Supplementary Table 5.

For constructing the plasmid for multiple genetic circuits, the ORF of each Cas protein was amplified using appropriate primers. Using each switch plasmid as a template, plasmid backbones were amplified by inverse PCR. The ORFs were inserted into the plasmid backbone using the In-Fusion HD Cloning Kit. Links to sequences of all the plasmids generated are listed in Supplementary Table 5.

All Plasmids were purified using the Midiprep kit (QIAGEN, #12143/12145 or Promega, #A2492/A2495).

### Construction of IVT (in vitro transcription) template

The ORF of SpCas9 (for trigger mRNA), 5′-UTR fragment, and 3′-UTR fragment were amplified by PCR using appropriate primers and template plasmids or oligo DNAs. IVT templates of trigger mRNA were generated by fusion PCR using the ORF, 5′-UTR fragment, 3′-UTR fragment, and appropriate primers.

To make an IVT template of switch mRNA, a fragment including the 5′-UTR and ORF was amplified from pGluc-Sp_gRNA-EGFP by PCR using appropriate primers. This and the 3′-UTR fragments were fused by fusion PCR using appropriate primers. IVT templates of EGFP (for control mRNA), iRFP670 (for reference mRNA), E3, K3, and B18R mRNAs were amplified with appropriate plasmids and primers by PCR.

To remove the template plasmids in the PCR products, 1 μL DpnI was added to each reaction, and the mixture was incubated at 37 °C for 30 min. All PCR products were purified using the MinElute PCR purification kit (QIAGEN, #28004) or Monarch PCR & DNA Cleanup Kit (NEB, #T1030) according to the manufacturer's protocols. The set of primers used is listed in Supplementary Table 6.

### mRNA synthesis and purification

All mRNAs were prepared using a MEGAscript T7 Transcription Kit (Thermo Fisher Scientific, #AM1333/AM1334). To reduce immune response, pseudouridine-5′-triphosphate (ΨTP) and 5-methylcytidine-5′-triphosphate (m5CTP) (both from TriLink BioTechnologies, #N-1019 and #N-1014, respectively) were used instead of natural UTP and CTP, respectively for trigger and reference mRNAs. For E3, K3, and B18R mRNAs, N1-methylpseudouridine (m1Ψ) (TriLink BioTechnologies, #N-1081) was used instead of natural UTP. Because modified bases affect the interaction of Cas protein and corresponding crRNA, natural rNTPs were used for the switch and control mRNAs. The IVT reaction included 1 × Enzyme mix, 1 × Reaction buffer, 7.5 mM ΨTP, m1Ψ or UTP, 7.5 mM m5CTP or CTP, 7.5 mM ATP, 1.5 mM GTP, 6 mM Anti Reverse Cap Analog (TriLink BioTechnologies, #N-7003), and the template DNA. The mixture was incubated at 37 °C for 6 h. To remove the template DNA, TURBO DNase (Thermo Fisher Scientific, included in a

MEGAscript T7 Transcription Kit) was added to the mixture and incubated at 37 °C for 30 min. Then the reaction mixtures were purified using a FavorPrep Blood/Cultured Cells total RNA extraction column (Favorgen Biotech, #FARBC-C50) or Monarch RNA Cleanup kit (NEB, #T2040) and incubated with Antarctic Phosphatase (NEB, #M0289) at 37 °C for 30 min. The reaction mixtures were purified again using the RNeasy MinElute Cleanup Kit (QIAGEN, #74204) or Monarch RNA Cleanup kit according to the manufacturer's protocols.

All IVT mRNA sequences are shown in Supplementary Sequences.

## Cell culture
HEK293FT cells (Invitrogen, #R70007) were cultured in DMEM High glucose (Nacalai Tesque, #08459-64) supplemented with 10% FBS (Biosera, #FB-1285/500), 2 mM L-Glutamine (Invitrogen, #25030-081), 0.1 mM Non-Essential Amino Acids (Invitrogen, #11140-050), and 1 mM Sodium Pyruvate (Sigma, #S8636). In Supplementary Fig. 17, HEK293FT cells that constitutively express TagBFP were used and cultured in the same manner as HEK293FT cells. HeLa (ATCC, #CCL-2) and A549 cells (RIKEN, #RCB3677) were cultured in DMEM High glucose supplemented with 10% FBS.

iPS cells (1383D6 (RIKEN BRC #HPS1006), EP004-2-57 dCas9) were cultured in StemFit AK02N (Ajinomoto, #AK02) on laminin-511 E8 (iMatrix-511 silk, Matrixome, #892021). All cell lines were cultured at 37 °C with 5% $CO_2$ in a humidified cell culture incubator. Doxycycline (Nacalai tesque, #19088-71) was added 4 h before the transfection (final concentration: 1 μg/mL).

## Generation of Tet-ON dCas9 iPS cells
Human 1383D6 iPS cells (RIKEN BRC, #HPS1006) were electroporated with a combination of an AAVS1 donor vector (3 μg, KW1368) and an AAVS1 TALEN pair (1 μg each) (KW1295_pCAG-TALdNC-AAVS1-T1-pA, Addgene, #80495; KW1296_pCAG-TALdNC-AAVS1-T2-pA, Addgene, #80495)[70]. Electroporation was performed using a NEPA21 electroporator (Nepa Gene Co. Ltd) with the following condition: Poring pulse (voltage: 125 V, pulse length: 5 ms, pulse interval: 50 ms, number of pulses: 2, decay rate: 10%, polarity: +); Transfer pulse (voltage: 20 V, pulse length: 50 ms, pulse interval: 50 ms, number of pulses: 5, decay rate: 40%, polarity: +/−). Subsequently, $5 \times 10^5$ cells were seeded in a 6-cm dish in AK02N media containing CultureSure Y-27632 (Wako, #036-24023) and treated 48 h later with neomycin (G418 sulfate, 175 μg/mL, Merck, #345812) for eight consecutive days. Resistant cells were pooled and passaged into a single well of a 6-well plate. Single clones were isolated as Tet-ON dCas9 iPS cells (EP004-2-57 dCas9) and validated by genotyping PCR, Southern blotting, and mCherry expression upon Dox treatment (1 μg/mL).

## Plasmid transfection
HEK293FT cells were seeded in 24-well ($1.0 \times 10^5$ cells/well), 96-well ($2.0 \times 10^4$ cells/well), or 384-well ($2.5 \times 10^3$ cells/well) plates. Appropriate plasmids were transfected into the cells using Lipofectamine 2000 (Invitrogen, #11668027/11668019) according to the manufacturer's protocols. Details of the transfection conditions for each experiment are shown in Supplementary Table 7. An iRFP670 plasmid was used as a transfection control in all experiments except for those shown in Fig. 5, for which a TagBFP plasmid was used. For the experiments shown in Fig. 3C, D, A/C heterodimerizer (TaKaRa, #635067)-containing media (final concentration 500 nM) were prepared and replaced before the transfection.

## mRNA transfection
HEK293FT cells and A549 cells were seeded in 24-well plates ($1.0–2.0 \times 10^5$ cells/well). Human iPS cell lines and HeLa cells were seeded in 24-well plates ($5.0 \times 10^4$ cells/well). Appropriate mRNAs were transfected

into the cells using Lipofectamine MessengerMAX (Invitrogen, #LMRNA015/LMRNA008) according to the manufacturer's protocols. Details of the transfection conditions for each experiment are shown in Supplementary Table 7.

## Apoptosis assay
At 24 h following transfection, cells were collected. Briefly, the medium was collected in a 1.5 mL tube, and PBS was added to the cells and collected in the 1.5 mL tube. The cells were treated with Accumax (Nacalai tesque, #17087-54), incubated at 37 °C for 10 min, and then collected in the 1.5 mL tube. The cell suspension was centrifuged at $200 \times g$ at room temperature for 5 min. The supernatant was removed, and the remaining cell pellet was washed with PBS and centrifuged at $200 \times g$ at room temperature for 5 min. After aspirating the supernatant, cells were stained with 2.5 μL AnnexinV, Alexa Fluor 488 conjugate (Life Technologies, #A13201), and 0.5 μL SYTOX Red dead cell stain (ThermoFisher, #S34859) in 50 μL Annexin-binding buffer (Life Technologies, #V13241) for 15 min and treated with Annexin-binding buffer to dilute the cell suspension.

## Cell imaging
Before the flow cytometry measurements, cell images were captured using the Cytell Cell Imaging System Version 3.6.7.19 (GE Healthcare Life Sciences) or CQ1 confocal image cytometer Version 1.07.01.01 (Yokogawa Electric Corporation). To capture each fluorescent image, the following channels were used: Blue channel (Ex. 390 nm/Em. 430 nm) for TagBFP, Green channel (Ex. 473 nm/Em. 512.5 nm) for EGFP and hmAG1, Orange channel (Ex. 544 nm/Em. 588 nm) for TagRFP, and Red channel (Ex. 631 nm/Em. 702 nm) for iRFP670. Captured images were analyzed using ImageJ (NIH).

## Imaging analysis and calculations
TIFF files of the captured fluorescent cell images were analyzed using ImageJ (version 1.53). Background intensities were subtracted using the rolling ball algorithm/method before calculation. Cell area was defined by a reference (iRFP670 or, for experiments shown in Fig. 5, TagBFP)-positive region. The median value of the reporter/reference within the defined area was calculated and used for the analysis.

In the orthogonality heatmaps, intensities were calculated using the following formulas:

$$\text{Defined value (DV)} = \text{median value of the reporter/reference in each cell} \quad (1)$$

$$\text{Relative value (RV)} = (\text{DV of trigger} +)/(\text{DV of trigger} -) \quad (2)$$

$$\text{Normalized fold change} = (\text{RV})/(\text{RV of "No gRNA" sample}) \quad (3)$$

In Fig. 5, normalized fluorescence intensities were calculated using the following formulas:

$$\text{Normalized fluorescence intensity} = (\text{DV of each condition})/(\text{DV of control condition}) \quad (4)$$

Control condition of RFP (translational regulation) was [pSp_gRNA-RFP (+), pTRE-hmAG1 (+), gRNA (NT), dSpCas9-VPR (−)].

Control condition of hmAG1 (transcriptional regulation) was [pSp_gRNA-RFP (−), pTRE-hmAG1 (+), gRNA (TRE), dSpCas9-VPR (+)].

In Fig. 8, normalized fluorescence intensities were calculated using the following formulas:

$$\text{Normalized fluorescence intensity} = (\text{DV of each state})/(\text{DV of [0,0] state}) \quad (5)$$

## Calculation of orthogonality

Acquired fluorescence intensity was calculated as normalized fold change (FC) with the above formula. As an evaluation of orthogonality, cosine distances of a combination of indexed Cas proteins and corresponding switches $C \subset \{1,2,..\}$ were calculated as follows:

$$cosine\ distance(C) = \min_{i,j \in C} cosine\ disntance(\mathbf{FC}(i|C), \mathbf{FC}(j|C)) \quad (6)$$

$$cosine\ distance(\mathbf{FC}(i|C), \mathbf{FC}(j|C)) = \frac{(\mathbf{FC}(i|C) - \mathbf{1}) \cdot (\mathbf{FC}(i|C) - \mathbf{1})}{|\mathbf{FC}(i|C) - \mathbf{1}| \cdot |\mathbf{FC}(i|C) - \mathbf{1}|} \quad (7)$$

$\mathbf{FC}(j|C)$ is a vector composed of Cas protein $j$'s fold change profile against switches in $C$. To obtain the best combinations of Cas proteins, the cosine distances of all the combinations under the number of selected proteins were calculated using a custom Python3 script.

## Flow cytometry measurements

The cells were washed with PBS (Nacalai tesque, #4249-24), treated with 0.25% Trypsin-EDTA (Thermo Fisher Scientific, #25200072) and incubated at 37 °C with 5% $CO_2$ for 5 min. The cells were analyzed using an Accuri C6 (BD Bioscience) flow cytometer with FL1 (533/30 nm) and FL4 (675/25 nm) filters. Accuri C6 was manipulated wih CFlow Plus Software (Version 1.0.227.4) or CSampler Software (Version 1.0.264.21).

## Flow cytometry data analysis

Flow cytometry data sets were analyzed using FlowJo version 10.5.3 (BD Biosciences) and Excel (Microsoft). Gates were generated using mock samples. Data from debris were eliminated when preparing forward- versus side-scatter dot plots (FSC-A versus SSC-A). Then, events on the chart edges in the dot plots of the EGFP intensity versus the iRFP670 intensity were removed. In the histogram where iRFP670-intensity is displayed on the X-axis, the iRFP670-positive (reference-positive) gate was defined (Supplementary Fig. 1C). In the following analysis, the median of reporter/reference of each cell was calculated from the reference positive population using FlowJo.

Relative reporter expression was defined using the following formulas:

$$Normalized\ intensity\ (NI) = 1000 \times$$
$$median\ of\ the\ ratio(reporter\ intensity/reference\ intensity)\ of\ each\ cell \quad (8)$$

$$Relative\ intensity\ (RI) = (NI\ of\ trigger+)/(NI\ of\ trigger-) \quad (9)$$

$$Relative\ reporter\ expression = (RI)/(RI\ of\ "No\ gRNA"\ sample) \quad (10)$$

In Supplementary Fig. 2, all RI values were normalized by the value of the [Trigger -] [No gRNA] sample.

In Fig. 7D, fold activation was defined as RI value normalized by the value of the [-,-,-] sample.

The fold change in Supplementary Fig. 17C was defined as RI in ON switch and the reciprocal of RI in OFF switch.

In Fig. 6D, the net fold-change was calculated by dividing the NI in ON state ([1,1]) by the averaged value of NI in OFF states ([0,0], [0,1], and [1,0]). Vector proximity angle is the angle between two 4-dimensional vectors. One is a truth table vector that has ideal output (= 0 or 1) for each state ([0,0], [1,0], [0,1], and [1,1]). The other is a vector that carries the observed output levels (=NI) of each state ([0,0], [1,0], [0,1], and [1,1]). Thus, $\theta$ ranges from 0° (best) to 90° (worst).

## Statistical analysis

All statistical analyses were performed by using R software or Excel (Microsoft).

## Multiple sequence alignment for phylogenetic tree

Representative Cas9 sgRNA sequences that showed crosstalk were analyzed with Multiple Sequence Comparison by Log- Expectation (MUSCLE: https://www.ebi.ac.uk/Tools/msa/muscle/). The result was exported as a FASTA file, and a phylogenetic tree was drawn by Python3.

## Reporting summary

Further information on research design is available in the Nature Portfolio Reporting Summary linked to this article.

# Data availability

All the data supporting this study are available within the main text and the Supplementary Information file. Source data are provided online with this paper.

# Code availability

Custom codes used in this study are available from GitHub. For image quantifications: https://zenodo.org/badge/latestdoi/428991142. For MFE calculation and orthogonality validation: https://zenodo.org/badge/latestdoi/578403793.

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

## Acknowledgements

We thank Dr. Peter Karagiannis, Dr. Kelvin K. Hui (Kyoto University), and Dr. Zoher Gueroui (École Normale Supérieure) for critical reading of the manuscript; Yusuke Shiba, Ryo Hirayama, and Shintaro Oe (Kyoto University) for helping with the experiments; and Miho Nishimura, Yuko Kono, Hiromi Takemoto, and Shodai Komatsu (Kyoto University) for their administrative support. We also thank Dr. Hideyuki Nakanishi (Tokyo Medical and Dental University) for the intein information, Dr. Yoshihiko Fujita (Kyoto University) for establishing the imaging quantification method, Dr. Akitsu Hotta (Kyoto University) for providing the AsCas12a ORF, and Shigetoshi Kameda (Kyoto University) for providing some IVT templates. This work was supported by JSPS KAKENHI Grant Number JP 19K16110 (to S.K.), 19J21199 (to H.O.), 20K15777 (to M.H.), 15H05722, and 20H05626 (to H.S.), and the iPS Cell Research Fund.

## Author contributions

S.K. and H.S. managed the project. S.K. conceived the idea. M.H. found the translational repression ability of SpCas9. S.K., H.O., M.H., and H.S. designed the project. S.K., H.O., M.H., and T.K. performed the experiments and analyzed the data. S.S. performed bioinformatic analysis, including calculation of MFE and the orthogonality verification. S.L. and K.W. established the Dox-inducible dCas9 cell line. H.O. mainly prepared the figures under the supervision of S.K.. S.K., H.O., M.H., and H.S. wrote the manuscript. S.K., H.O., and M.H. contributed equally to this work.

## Competing interests

Kyoto University has filed a patent application regarding the CARTRIDGE method (JP2020044905). S.K., H.O., M.H., and H.S. are the inventors of record listed on the patents. H.S. own shares of aceRNA Technologies Ltd., and has outside director of aceRNA Technologies Ltd. The other authors declare no competing interests.
