## [Peer Review File · Nature Communications]

Reviewers' Comments:

Reviewer #1:

Remarks to the Author:

Kawasaki and Saito et al. reported a CRISPR-based translation regulation system CaTRIDGE and demonstrated it works as an OFF switch. Due to the existing repertoire of Cas9 proteins, they could also establish many orthogonal OFF switches, which was impressive. Applying these OFF switches to a NOT gate to a CRISPRa output, they also established a similar number of ON switches. I already liked the idea and appreciated the work done by the authors when it was initially submitted to Nature Chemical Biology. I, however, wondered about its scalability, and probably suggested too many experiments, but they performed an impressive number of experiments and cleared all of my questions and concerns. More comments follow:

(1) Supplementary Figs. 19 and 20 are beautiful. Why don't you put it as the main figures? I feel that Supplementary Fig. 19C and D and Supplementary Fig. 20 can go to Fig. 3, and Supplementary Fig. A and B can go to Fig. 7 to showcase more complex circuits.

(2) I know I was the one who suggested clarifying how the measured signals were normalized, but now I feel like having the acronyms NI and RI in the main text is too much, especially because these terms do not (need to) show later in the manuscript. Maybe just avoid these acronyms?

(3) Fig. 1D: Can they show mCherry signal data along with these conditions?

(4) Page 6, line 1: Were they all done by DNA plasmid transfection?

(5) Fig. 1E: it might be interesting to project the figure with a phylogenetic tree of Cas proteins and their sizes, and discuss the efficiencies along with these other features.

(6) Fig. 2A: What was the intron for?

(7) Page 8, line 8: What do you mean by "expression efficiency" here?

(8) Fig. 4 is great... I think it's totally ok to make it a bit busier by putting the contents of Supplementary Figs. 13 and 15. The science community would appreciate the quantitatively measured data.

Reviewer #2:

Remarks to the Author:

I thank the authors for responding thoroughly to the comments.

I still have one point that refers to the original comment on SF4 (the histogram representing EGFP/iRFP ratio). I still fail to understand what it represents. Also, to plot it as a histogram (thus a distribution), it means that they calculated the ratio of EGFP/iRFP for each cell (dot)? How was this calculation carried out?

Linked to the response of the authors to this original comment is the following:

-Fig 1: After the normalization on plus/- Cas9, the authors perform a further normalization using a EGFP that lacks the gRNA (no gRNA). So, the level of repression is ultimately calculated on the

EGFP that does not contain any element in the 5'UTR.

It seems that this further layer has been added in response referee #1 to account for the metabolic load due to addition of spCas9. However, it has been observed that sometimes addition of sequences in the 5'UTR of a gene can affect the expression of the resulting protein regardless of additional regulators (although it does not seem the case from the fluorescent images in SF2). It would be beneficial to see the absolute EGFP levels (prior normalization) in the no gRNA and gRNA ON (no Cas9) condition. If there is a constructive reason to carry out this normalization, a more appropriate control would then be a gRNA that harbors mutation such that Cas9 does not bind (although this experiment is not required).

My recommendation is to not normalize by the no gRNA control.

-Linked to the previous comment, in SF11 the authors rightly investigate the potential effect of sgRNAs on the mRNA translation. I suggest the authors add to the prediction based on the MFEs, also a plot of the EGFP fluorescence levels (au) for each of the sgRNAs used.

-In the bar plots representing the Relative reporter expression the a.u. (absolute units) should be replaced by r.u. (relative units)

Reviewer #3:

Remarks to the Author:

In the manuscript by Kawasaki et al, the authors built a new system for repurposing Cas proteins as translational modulators (CARTRIDGE) and verified the characteristics a large amount of translation OFF and ON switches. By combining CARTRIDGE with other CRISPR-related technologies such as CRISPRa, split-Cas9 and anti-CRISPR, they also constructed a variety of artificial circuits. In my opinion, the idea of CARTRIDGE is novel and experiments were performed to demonstrate the practical application of this tool. I suggest to publish this research in Nature Communications after minor modification. My comments are shown below:

1. It would be more accessible to readers if the authors can explain how the protospacer sequence of gRNA was selected in CaRTRIDGE in the main text.
2. Page 7, line 1: The authors speculate the weakest performance of PspCas13b as an ON switch is due to the endoribonucleolytic activities. Can the authors explain in detail? Why the pre-crRNA cleaving activity may cause a poor performance? In addition, the authors didn't provide direct evidence. If possible, it would be great to test the dependency of ON switches on endoribonucleolytic activities of PspCas13b/PguCas13b.
3. Page 9, line2: DnaE instead of DnaB?
4. Figure 3C: I wonder why there is a decreased relative reporter expression when only the N-Cas9 is present. Is it because N-Cas9 has a slight sgRNA/crRNA binding activity? Or other reasons?
5. The authors tested the performance of OFF switch in 4 cell types (Figure 1C). I wonder which cell types were used for the remaining circuits (e.g. split-Cas9-based NAND gate)? It would be better to claim the cell type in the captions.

Reviewer #1 (Remarks to the Author):

In the manuscript by Kawasaki et al, the authors built a new system for repurposing Cas proteins as translational modulators (CARTRIDGE) and verified the characteristics a large amount of translation OFF and ON switches. By combining CARTRIDGE with other CRISPR-related technologies such as CRISPRa, split-Cas9 and anti-CRISPR, they also constructed a variety of artificial circuits. In my opinion, the idea of CARTRIDGE is novel and experiments were performed to demonstrate the practical application of this tool. I suggest to publish this research in Nature Communications after minor modification. My comments are shown below:

Response:

We thank the referee again for the positive feedback and valuable comments on how we could improve our study and manuscript. We provided a point-by-point response to the reviewer's comments as follows.

1. It would be more accessible to readers if the authors can explain how the protospacer sequence of gRNA was selected in CaRTRIDGE in the main text.

Response:

We thank the referee for this suggestion. We added the text below into the main text:

Page 6, line 18,

*“Several Cas-responsive switches contained spacers complementary to an exogenous gene, *Gaussia luciferase*, but we did not observe any clear indications that a spacer sequence was necessary (Supplementary Table 2 and Supplementary Figure 7).”*

2. Page 7, line 1: The authors speculate the weakest performance of PspCas13b as an ON switch is due to the endoribonucleolytic activities. Can the authors explain in detail? Why the pre-crRNA cleaving activity may cause a poor performance? In addition, the authors didn't provide direct evidence. If possible, it would be great to test the dependency of ON switches on endoribonucleolytic activities of PspCas13b/PguCas13b.

[Redacted]

3. Page 9, line2: DnaE instead of DnaB?

Response:

We thank the referee for pointing this out. We corrected DnaE to DnaB in the main text and Figure 3A.

4. Figure 3C: I wonder why there is a decreased relative reporter expression when only the N-Cas9 is present. Is it because N-Cas9 has a slight sgRNA/crRNA binding activity? Or other reasons?

Response:

We thank the referee for this comment. We speculate that N-Cas9 alone can weakly interact with sgRNAs in mammalian cells. This tendency was observed not only in Figure 3B but also in Supplementary Figure 12.

5. The authors tested the performance of OFF switch in 4 cell types (Figure 1C). I wonder which cell types were used for the remaining circuits (e.g. split-Cas9-based NAND gate)? It would be better to claim the cell type in the captions.

Response:

We agree with the referee on this suggestion. In the new figure legends, we described the cell lines used in each experiment.

Reviewer #2 (Remarks to the Author):

I thank the authors for responding thoroughly to the comments.

Response:

We thank the referee again for the valuable comments and for taking the time to read our manuscript closely. We provided a point-by-point response to the reviewer's comments as follows.

I still have one point that refers to the original comment on SF4 (the histogram representing EGFP/iRFP ratio). I still fail to understand what it represents. Also, to plot it as a histogram (thus a distribution), it means that they calculated the ratio of EGFP/iRFP for each cell (dot)? How was this calculation carried out?

Response:

We thank the referee for the comment. To normalize the transfection efficiency for each experiment, we always transfected a reference plasmid that encodes iRFP670 with the switch plasmid that encodes EGFP. By calculating the EGFP/iRFP670 ratio, we can estimate translation ability independent of transfection efficiency. Similar analyses have been performed in previous studies (e.g., PMID: 26004781, 31765565, and 34985961).

We did indeed calculate the EGFP/iRFP670 ratio for each cell (dot). To do this, we used FlowJo, a software for flow cytometry data analysis that has a function to calculate the ratio between any two collected parameters for each cell (dot).

We revised the text below into the main text:

Page 4, line 32,

“To analyze the translational repression ability of SpCas9, we first normalized the reporter expression level (EGFP) by the expression level of our reference protein (iRFP670) to account for variations in transfection efficiency (Normalized Intensity, see Materials and Methods section).”

Linked to the response of the authors to this original comment is the following:

-Fig 1: After the normalization on plus/- Cas9, the authors perform a further normalization using a EGFP that lacks the gRNA (no gRNA). So, the level of repression is ultimately calculated on the EGFP that does not contain any element in the 5'UTR.

It seems that this further layer has been added in response referee #1 to account for the metabolic load due to addition of spCas9. However, it has been observed that sometimes addition of sequences in the 5'UTR of a gene can affect the expression of the resulting protein regardless of additional regulators (although it does not seem the case from the fluorescent images in SF2). It would be beneficial to see the absolute EGFP levels (prior normalization) in the no gRNA and gRNA ON (no Cas9) condition. If there is a constructive reason to carry out this normalization, a more appropriate control would then be a gRNA that harbors mutation such that Cas9 does not bind (although this experiment is not required).

My recommendation is to not normalize by the no gRNA control.

Response:

We thank the referee for this insightful comment and for allowing us to reconsider our data. We reanalyzed the data as suggested and did not observe any meaningful differences between comparisons using relative and absolute reporter intensities. We added these data as Supplementary Figure 1E (please see the right panel of the figure below).

As for “a gRNA that harbors mutation such that Cas9 does not bind,” it is difficult to prepare all possible mutants of such gRNAs. Only a few RNA sequences have been verified to disrupt the binding of cognate Cas proteins. The reason for the need to use a control reporter construct is to eliminate the possibility that the expression level of the reporter could be altered by Cas protein induction itself. “No gRNA” construct satisfies the following conditions: (1) a control construct should not bind to Cas proteins, and (2) suitable for eliminating the effect of Cas protein induction itself; therefore, we believe it to serve as an appropriate control.

The normalization reduces the possibility that changes in reporter expression by the gRNA-harboring switches were caused by factors such as Cas protein induction itself, transfection efficiencies, or the insertion of Cas-binding motifs.

-Linked to the previous comment, in SF11 the authors rightly investigate the potential effect of sgRNAs on the mRNA translation. I suggest the authors add to the prediction based on the MFEs, also a plot of the EGFP fluorescence levels (au) for each of the sgRNAs used.

Response:

As we mentioned in our responses to the two referee comments above, we normalized the EGFP intensity by the reference iRFP670 intensity (EGFP/iRFP670 ratio) to account for variations in transfection efficiency and observed similar trends in reporter expression levels, with or without normalization (new Supplementary Figure 1E and Supplementary Figure 2). Therefore, we displayed the plots based on the normalized values (e.g., MFE vs ON state expression level) in Supplementary Figure 11.

-In the bar plots representing the Relative reporter expression the a.u. (absolute units) should be replaced by r.u. (relative units)

Response:

We thank the referee for this comment. We abbreviated “arbitrary unit” as a.u. To clarify this point, we added an explanation of this in all the figure legends.

Reviewer #3 (Remarks to the Author):

Kawasaki and Saito et al. reported a CIRSPR-based translation regulation system CaRTRIDGE and demonstrated it works as an OFF switch. Due to the existing repertoire of Cas9 proteins, they could also establish many orthogonal OFF switches, which was impressive. Applying these OFF switches to a NOT gate to a CRISPRa output, they also established a similar number of ON switches. I already liked the idea and appreciated the work done by the authors when it was initially submitted to Nature Chemical Biology. I, however, wondered about its scalability, and probably suggested too many experiments, but they performed an impressive number of experiments and cleared all of my questions and concerns. More comments follow:

Response:

We thank the referee again for the positive feedback and valuable comments on how we could improve our study and manuscript. We provided a point-by-point response to the reviewer's comments as follows.

(1) Supplementary Figs. 19 and 20 are beautiful. Why don't you put it as the main figures? I feel that Supplementary Fig. 19C and D and Supplementary Fig. 20 can go to Fig. 3, and Supplementary Fig. A and B can go to Fig. 7 to showcase more complex circuits.

Response:

We thank the referee for this wonderful suggestion. It was difficult to make each figure fit on an A4 page if Supplementary Figures 19 and 20 were placed together with Figure 3 and the previous Figure 7. Instead, we made Supplementary Figures 19 and 20 into the new Figure 7.

(2) I know I was the one who suggested clarifying how the measured signals were normalized, but now I feel like having the acronyms NI and RI in the main text is too much, especially because these terms do not (need to) show later in the manuscript. Maybe just avoid these acronyms?

Response:

We thank the referee for the suggestion. We removed the acronyms NI and RI in the main text.

(3) Fig. 1D: Can they show mCherry signal data along with these conditions?

Response:

We thank the referee for this comment. As suggested, we added the representative fluorescence microscopy images of mCherry in Supplementary Figure 5.

(4) Page 6, line 1: Were they all done by DNA plasmid transfection?

Response:

Yes, we used plasmids to investigate the performance of all those switches.

(5) Fig. 1E: it might be interesting to project the figure with a phylogenetic tree of Cas proteins and their sizes, and discuss the efficiencies along with these other features.

Response:

We thank the referee for this insightful comment. We agree that it would be interesting to examine further the relationship between translational regulatory efficiencies and other features like the evolutionary distance between species (or protein sequence similarity). We speculate that further studies will clarify this point by validating the performance of switches with Cas proteins from related species. On the other hand, we could not observe a clear relationship between switch performance and Cas protein size. We suspect that there are additional parameters that determine switch performance.

(6) Fig. 2A: What was the intron for?

Response:

An intron is required to activate NMD artificially on the ON switches. During the pioneer round of translation, a translating ribosome displaces exon junction complexes (EJCs), which are multi-protein complexes deposited on spliced mRNAs. If premature termination codons (PTCs) are present, downstream EJCs are not removed, and NMD is triggered. The intron was included so that splicing would occur on the switch mRNA and interacts with EJCs.

(7) Page 8, line 8: What do you mean by “expression efficiency” here?

Response:

On page 8, line 8, we described “repression efficiencies.” As shown in Supplementary Figure 11, “repression efficiency of X%” means that the relative reporter expression of a switch was calculated as X% less than the reporter expression of the “No gRNA” control.

(8) Fig. 4 is great... I think it's totally ok to make it a bit busier by putting the contents of Supplementary Figs. 13 and 15. The science community would appreciate the quantitatively measured data.

Response:

We thank the referee for this suggestion. We modified Figure 4 to include the quantitative data from Supplementary Figures 15A and C.